# Decoys provide a scalable platform for the identification of plant E3 ubiquitin ligases that regulate circadian function

**Ann Feke[1], Wei Liu[1†], Jing Hong[1,2†], Man-Wah Li[1], Chin-Mei Lee[1], Elton K Zhou[1], Joshua M Gendron[1]\***

[1]Department of Molecular, Cellular and Developmental Biology, Yale University, New Haven, United States; [2]School of Food Science and Engineering, South China University of Technology, Guangzhou, China

**Abstract** The circadian clock relies on regulated degradation of clock proteins to maintain rhythmicity. Despite this, we know few components that mediate protein degradation. This is due to high levels of functional redundancy within plant E3 ubiquitin ligase families. In order to overcome this issue and discover E3 ubiquitin ligases that control circadian function, we generated a library of transgenic Arabidopsis plants expressing dominant-negative 'decoy' E3 ubiquitin ligases. We determined their effects on the circadian clock and identified dozens of new potential regulators of circadian function. To demonstrate the potency of the decoy screening methodology to overcome redundancy and identify *bona fide* clock regulators, we performed follow-up studies on *MAC3A* (*PUB59*) and *MAC3B* (*PUB60*). We show that they redundantly control circadian period by regulating splicing. This work demonstrates the viability of ubiquitin ligase decoys as a screening platform to overcome genetic challenges and discover E3 ubiquitin ligases that regulate plant development.
DOI: https://doi.org/10.7554/eLife.44558.001

**\*For correspondence:**
joshua.gendron@yale.edu

[†]These authors contributed equally to this work

**Competing interests:** The authors declare that no competing interests exist.

## Introduction

The circadian clock is essential for proper coordination of biological processes with the environment. In plants, the circadian clock controls diverse aspects of plant development, including hypocotyl elongation, leaf movement, seasonal flowering time, and stress responses, both biotic and abiotic (*Dowson-Day and Millar, 1999*; *Fowler et al., 1999*; *Hoshizaki and Hamner, 1964*; *Ingle and Roden, 2014*; *Liu et al., 2013*; *Nakamichi et al., 2016*). The timing of the clock is set by environmental inputs, such as daily changes in light and temperature, but it is also self-sustaining and capable of maintaining roughly 24 hr rhythms in the absence of changes in the environmental signals. These self-sustaining oscillations are driven by interlocking transcriptional feedback loops that result in successive expression of a series of transcriptional repressors and activators throughout the day (*McClung, 2014*; *Ronald and Davis, 2017*).

In plants, the transcriptional repressors consist predominantly of three groups of proteins. These are the morning expressed CIRCADIAN CLOCK ASSOCIATED 1 (CCA1) and LATE ELONGATED HYPOCOTYL (LHY), the morning and afternoon expressed PSEUDO-RESPONSE REGULATOR (PRR) family, and the evening expressed evening complex (including EARLY FLOWERING 3 (ELF3), EARLY FLOWERING 4 (ELF4) and LUX ARRYTHMO (LUX)) (*Alabadí et al., 2001*; *Alabadí et al., 2002*; *Carré and Kim, 2002*; *Doyle et al., 2002*; *Farré et al., 2005*; *Fujimori et al., 2005*; *Gendron et al., 2012*; *Hazen et al., 2005*; *Helfer et al., 2011*; *Hicks et al., 1996*; *Hicks et al., 2001*; *Kikis et al., 2005*; *Mizoguchi et al., 2002*; *Nakamichi et al., 2005b*; *Nakamichi et al., 2005a*; *Onai and Ishiura, 2005*; *Schaffer et al., 1998*; *Wang and Tobin, 1998*). More recently, the LIGHT-REGULATED WD

**eLife digest** Plants have an internal time keeper known as the circadian clock that operates in 24-hour cycles to coordinate the plants behaviors with the environment. The clock is made of many different proteins and plants carefully control when they make and destroy these proteins to regulate the cycle.

Inside plant cells, enzymes known as E3 ubiquitin ligases determine which proteins are destroyed by labelling target proteins with a small tag. Plants have hundreds of different E3 ubiquitin ligases, leading to overlaps in the roles the different enzymes play. These overlaps make it difficult to identify the specific E3 ubiquitin ligases that are involved in a particular process. As a result, only few E3 ubiquitin ligases implicated in the circadian clock have been identified so far.

A small weed known as *Arabidopsis* is often used in research studies because it grows quickly and the genes can be easily manipulated. Here, Feke et al. set out to develop a new tool to identify the specific E3 ubiquitin ligases involved in regulating the circadian clock in *Arabidopsis*.

The team created a library of hundreds of *Arabidopsis* plants producing different decoy E3 ubiquitin ligases that retained their ability to bind to target proteins but were unable to degrade them. Nearly a quarter of the E3 ligases found in *Arabidopsis* were represented in this library. The decoy enzymes protected the target proteins from being degraded by the normal E3 ubiquitin ligases, resulting in the library plants having presumably higher levels of these target proteins compared to normal *Arabidopsis* plants. By tracking circadian rhythms in these plants, the team was able to identify the individual E3 ligases that control the circadian clock.

The experiments revealed several E3 ligases that may regulate the circadian clock, including two enzymes called MAC3A and MAC3B. Further experiments demonstrated that MAC3A and MAC3B have similar roles in regulating the circadian clock and can compensate for the absence of the other.

The library of *Arabidopsis* plants generated by Feke et al. is now available for other researchers to use in their studies. In the future this approach could be adapted to make similar libraries for crops and other plants that have even more E3 ligase enzymes than *Arabidopsis*.

DOI: https://doi.org/10.7554/eLife.44558.002

(LWD), REVEILLE (RVE), and NIGHT LIGHT–INDUCIBLE AND CLOCK-REGULATED1 (LNK) genes, were identified as critical transcriptional activators in the plant clock and provide a more comprehensive understanding of the transcriptional feedback loops that drive oscillations (*Farinas and Mas, 2011*; *Hsu et al., 2013*; *Rawat et al., 2011*; *Wu et al., 2016*; *Xie et al., 2014*).

Eukaryotic circadian clocks employ the ubiquitin proteasome system (UPS) to degrade clock transcription factors at the appropriate time of day (*Grima et al., 2002*; *He et al., 2003*; *Ito et al., 2012*; *Ko et al., 2002*; *Shirogane et al., 2005*). The UPS is ideally suited for regulation of the circadian clock because it can mediate protein degradation quickly and specifically. To achieve specificity, the UPS leverages E3 ubiquitin ligase proteins (*Chen and Hellmann, 2013*; *Hua and Vierstra, 2011*). E3 ubiquitin ligases act as substrate adaptor proteins by bringing the substrate into proximity of an E2 ubiquitin conjugating enzyme to promote substrate ubiquitylation. Once a lysine-48-linked poly-ubiquitin chain is added to the substrate, it is sent to the proteasome where it is degraded (*Vierstra, 2009*). In addition to their role in the ubiquitin proteasome system, E3 ubiquitin ligases also coordinate ubiquitylation that regulates other processes, such as endocytosis or the formation of protein complexes (*Komander and Rape, 2012*). E3 ubiquitin ligases exist in multiple families and contain highly diverse protein recognition domains, allowing them to achieve specificity in the system.

F-box proteins are the substrate adaptor component of a larger E3 ubiquitin ligase complex and are utilized by all eukaryotic circadian clocks (*Grima et al., 2002*; *He et al., 2003*; *Ito et al., 2012*; *Ko et al., 2002*; *Shirogane et al., 2005*). The complex, abbreviated SCF, consists of S PHASE KINASE-ASSOCIATED PROTEIN 1 (SKP1), CULLIN, RING-BOX1 (RBX1), and the F-box protein (*Bai et al., 1996*; *Deshaies, 1999*; *Deshaies and Joazeiro, 2009*; *Hua and Vierstra, 2011*; *Lechner et al., 2006*). A family of three partially redundant F-box proteins, ZEITLUPE (ZTL), LOV KELCH PROTEIN 2 (LKP2), and FLAVIN-BINDING KELCH REPEAT 1 (FKF1), regulate the circadian clock and flowering time in plants (*Imaizumi et al., 2005*; *Imaizumi et al., 2003*; *Nelson et al.,*

*2000*; *Schultz et al., 2001*; *Somers et al., 2000*). ZTL, which has the largest impact on clock function, regulates stability of TOC1, PRR5, and CHE (*Fujiwara et al., 2008*; *Kiba et al., 2007*; *Lee et al., 2018*; *Más et al., 2003*). Outside of the ZTL family, some evidence suggests that LHY stability is regulated by the non-F-box RING-type E3 ubiquitin ligase SINAT5 (*Park et al., 2010*). Since the discovery of these E3 ubiquitin ligases, little progress has been made in identifying additional E3 ubiquitin ligases that participate in clock function.

The inability to identify plant E3 ubiquitin ligases that regulate the circadian clock is likely due to genetic challenges that hamper traditional forward genetic approaches. In Arabidopsis, gene duplication has led to expansion of the genes involved in UPS function (*Navarro-Quezada et al., 2013*; *Risseeuw et al., 2003*; *Yee and Goring, 2009*). For instance, there are approximately 700 Arabidopsis F-box genes, while in humans there are 69 (*Finn et al., 2016*; *Grima et al., 2002*; *Kuroda et al., 2002*; *Xu et al., 2009*). This has likely led to increased functional redundancy rendering gene knockouts an inefficient method to identify function. To support this, the majority of *ztl* mutant alleles are semi-dominant (*Kevei et al., 2006*; *Martin-Tryon et al., 2007*; *Somers et al., 2004*; *Somers et al., 2000*). This suggests that reverse genetic strategies may be a more potent approach to identify E3 ubiquitin ligases that regulate clock function.

In order to overcome redundancy in plant E3 ubiquitin ligase families, we developed a 'decoy' E3 ubiquitin ligase approach. The decoy approach involves expressing an E3 ubiquitin ligase that lacks the ability to recruit the E2 conjugating enzyme but retains the ability to bind to the substrate (*Han et al., 2004*; *Kishi and Yamao, 1998*; *Latres et al., 1999*; *Li et al., 2012*; *Zhou et al., 2015*). We have shown that this inactivates the full-length E3 ubiquitin ligase and acts to stabilize the substrate protein (*Lee et al., 2018*). The decoy acts as a dominant-negative, making it an effective genetic tool to identify the function of redundant E3 ubiquitin ligases. Additionally, the decoy stabilizes interaction with substrate proteins. This allows us to express the decoy with an affinity tag to study interactions between E3 ubiquitin ligases and substrates.

Here, we demonstrate the potency and scalability of the decoy technique by performing a reverse genetic screen to identify regulators of the circadian clock. We attempted to create decoy-expressing transgenic plants for half of the F-box-type E3 ubiquitin ligases and all of the U-box- type E3 ubiquitin ligases from Arabidopsis. Our completed library contains nearly ¼ of the Arabidopsis E3 ubiquitin ligases (*Vierstra, 2009*), spanning sixteen different protein-protein interaction domain classes, and including many genes with known functions as well as many that have not been studied in detail previously.

We used the decoy library to identify E3 ubiquitin ligases that can regulate the plant circadian clock. We uncovered a surprisingly large number of genes that regulate clock function with minor effects and a smaller number with more dramatic effects on clock period or phase. We then perform focused genetic studies on *PLANT U-BOX 59* and *PLANT U-BOX 60 (MAC3A* and *MAC3B)*, two homologous U-box genes which have been previously implicated in splicing. We go on to determine their molecular function in the clock by showing that the core clock gene, *PRR9*, is mis-spliced in the *mac3a/mac3b* double mutant. This work demonstrates the effectiveness of the decoy technique as a screening platform and identifies the first U-box-type E3 ligases that are involved in clock function in any system. It also establishes two important community resources: a list of E3 ligases that regulate the plant circadian clock, and a decoy library that is freely available and can be used to identify E3 ubiquitin ligases involved in any plant developmental processes.

## Results

### Construction of the decoy library

In order to discover E3 ubiquitin ligases that regulate the plant circadian clock, we created a library of transgenic plants expressing decoy E3 ubiquitin ligases. Decoy E3 ubiquitin ligases are identical to the native E3 ubiquitin ligases but lack the domain that recruits the E2 conjugating enzymes. Thus, the decoys retain substrate binding abilities but lack the ability to mediate substrate ubiquitylation (*Han et al., 2004*; *Kishi and Yamao, 1998*; *Latres et al., 1999*; *Lee et al., 2018*; *Zhou et al., 2015*). Thus, transgenic plants expressing decoy ubiquitin ligases should act dominantly to endogenous E3 ubiquitin ligases.

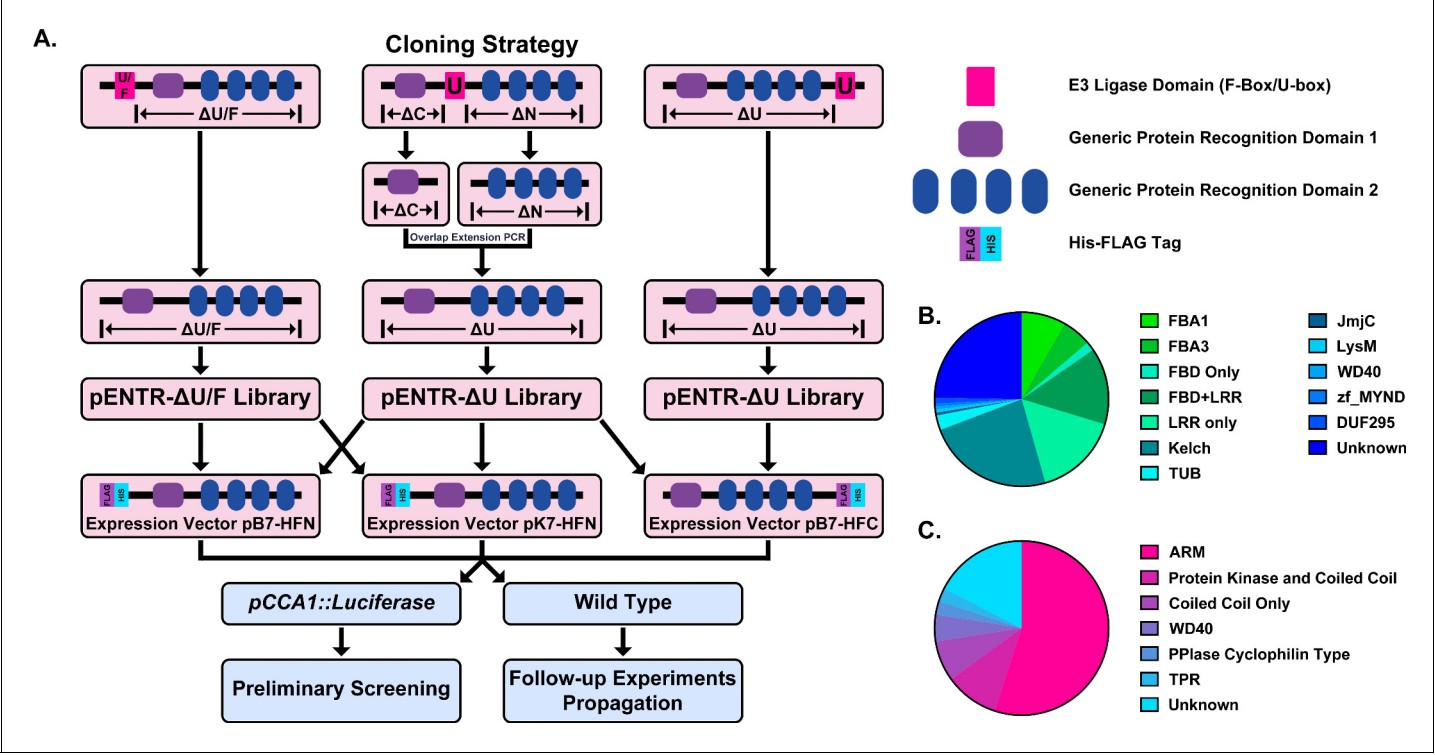

**Figure 1.** Construction of the Arabidopsis Decoy Library. (**A**) Cloning and experimental workflow of the F-box and U-box decoy library. F-box decoys follow the same path as the N-terminal U-box decoys. (**B**) Distribution of protein recognition domains in the F-box decoy library (**C**) Distribution of protein recognition domains in the U-box decoy library.

DOI: https://doi.org/10.7554/eLife.44558.003

In this pilot screen, we started with the F-box family of E3 ubiquitin ligases, in part because of the known role of F-box proteins on the circadian clock and in part because we have demonstrated the viability of the technique with three F-box E3 ubiquitin ligases (*Lee et al., 2018*). We selected roughly half of the F-box gene family, including representatives from all of the large classes and most of the small classes of F-box proteins for this initial screen. Some genes that we chose have differential expression under various growth conditions, while there was nothing known about the expression of others.

The F-box domain is unusual in that it is almost always located in the N-terminal portion of the protein. Thus, F-box decoy constructs were created by amplifying the sequence downstream of the F-box domain in each gene and creating pENTR vectors for each. The decoy constructs were then recombined into a vector that will drive their expression under a *CaMV* (*Cauliflower Mosaic Virus*) *35S* promoter and in-frame with a 6xHis/3xFLAG affinity tag (*Figure 1a*).

In order to test the decoy technique's viability across E3 ubiquitin ligase classes, we also selected a second family of E3 ubiquitin ligases to include in the library. The U-box family was selected due to its small size, containing only around 60 members (*Azevedo et al., 2001*; *Finn et al., 2016*; *Yee and Goring, 2009*). Furthermore, the U-box domain itself is well-defined and roughly the same size as the F-box domain (*Andersen et al., 2004*; *Aravind and Koonin, 2000*). Unlike the F-box domain, which is characteristically in the N-terminus of the protein, the U-box domain can be located anywhere throughout the protein sequence. For U-boxes with the U-box domain in the C- or the N-terminus, we amplified all sequence that was located upstream or downstream of the U-box, respectively. For those with the U-box domain located in the middle, we amplified both upstream and downstream sequences and then ligated the two halves together, a successful strategy that we utilized in our previous study (*Lee et al., 2018*). The decoy constructs were then recombined into the same expression vectors as described for the F-box decoy library (*Figure 1a*).

Based on another large scale cloning project in Arabidopsis, we expected 70–80% success rate in cloning the F-box genes (*Pruneda-Paz et al., 2014*). In fact, we were able to clone 82% of the attempted F-box genes, and ultimately succeeded in isolating transgenic plants for 65% of cloned decoy F-boxes (*Supplementary file 1*). The inability to isolate transgenic plants for the remaining 35% of cloned decoy F-boxes may be due to a multitude of factors, including but not limited to lethality caused by expressing the decoy, reduced transformation efficiency, or other technical constraints. Of those successfully generated transgenics, the majority contained either an LRR, Kelch, or F-box Associated (FBA1, FBA3, or FBD) protein recognition domain (at 30%, 24%, or 30%, respectively), with some F-box proteins containing both LRR and FBD domains together (14%) (*Figure 1b*). The remaining F-boxes contained a small number of other domains (6%), including TUBBY-like or WD40 domains, or no known protein recognition domain (25%). We also generated transgenic plants expressing 65% of the U-box family (*Supplementary file 1*). Of those cloned, 55% contained ARM repeats, 10% contained a Protein Kinase domain, 7.5% contained only a coiled coil region, 5% contained a WD repeat, 5% contained other annotated domains, and 17.5% contained no annotated domains (*Figure 1c*).

In sum, we attempted to generate a transgenic library expressing decoys for approximately 1/4th of the Arabidopsis E3 ubiquitin ligases (*Vierstra, 2009*). From here on, we use the term 'decoy' to describe a transgenic plant containing the *35S* promoter driven, FLAG-His tagged E3 ubiquitin ligase with the E3 ubiquitin ligase domain deleted. A decoy 'plant' is defined as a single, independent T1 insertion transgenic containing a decoy construct, and a decoy 'population' is a group of decoy plants which all express the same decoy transgene but are independent T1 transgenics.

## Screen design

In order to identify E3 ubiquitin ligases that regulate clock function, we transformed our decoy library into transgenic Col-0 plants harboring the *CIRCADIAN CLOCK ASSOCIATED* 1 promoter driving the expression of the *Luciferase* gene (*CCA1p::Luciferase*) and monitored clock function (*Pruneda-Paz et al., 2009*). From automated imaging experiments performed under constant light conditions on week-old seedlings entrained in LD (12 hr light/12 hr dark) conditions, we were able to measure clock period, phase, and relative amplitude error (RAE – a statistical measure of rhythmicity [*Moore et al., 2014*; *Zielinski et al., 2014*]) of all transgenic plants and controls.

As a quality control measure, we first filtered our data for those with reliable control experiments. For an experiment to be included in the analyses we required that the parental control *CCA1p::Luciferase* populations have a standard deviation of less than 0.75 hr. We chose this threshold value as it equates to the closest 15 min window to a 95% confidence interval within a 24 hr period. We removed any experiments with larger control variances from further analyses. By discarding datasets with larger degrees of variation, we reduce the chances of false positives and reduce the impact of any unpredictable environmental differences.

## The role of E3 ubiquitin ligase decoys in clock rhythmicity

Some circadian clock mutants completely ablate clock function and cause arrhythmicity (*Hazen et al., 2005*; *Nakamichi et al., 2009*). To determine the rhythmicity of the decoy plants we calculated the RAE for all 8502 individual T1 transgenic plants. We plotted each F-box and U-box gene from the screen on trees so that any potential redundant genes would be nearer to each other (*Dereeper et al., 2008*). The trees do not have evolutionary significance and only provide relative gene relatedness at the protein sequence level and were created using the full-length rather than decoy sequence. Five individual plants had an RAE greater than 0.6 which signifies lack of rhythmicity (*Figures 2–3*). In comparison, no control plants (n = 1783) had an RAE greater than 0.6 (*Figure 2—figure supplement 1* and *Figure 3—figure supplement 1*). No decoy populations had more than one arrhythmic plant, making it unlikely that any decoy ablates clock function. Rather, it is possible that the insertion landed in a gene necessary for rhythmicity in these plants.

## The effects of E3 ubiquitin ligase decoys on clock phase

We next determined whether the decoy populations have alterations in phasing of the Arabidopsis circadian clock. We calculated phase difference for each transgenic plant. This was done by calculating the average phase of the control population in each experiment, then subtracting this value from

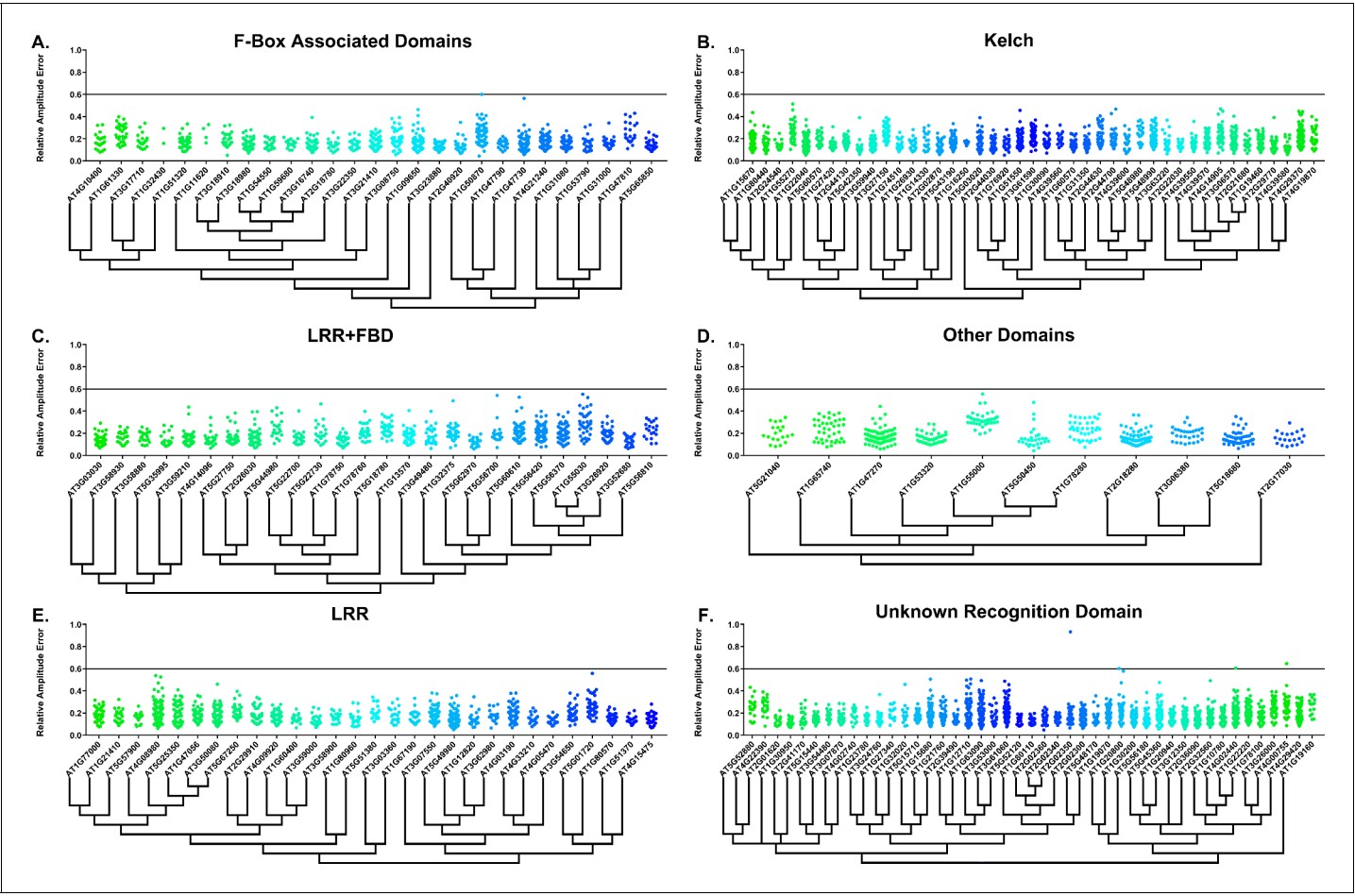

**Figure 2.** RAE distributions of F-box decoy plants. Values presented are the RAE for each individual T1 seedling. The black line represents the standard RAE cutoff of 0.6. Genes are separated by protein recognition domain and ordered by closest protein homology using Phylogeny.Fr, (*Dereeper et al., 2008*), and a tree showing that homology is displayed beneath the graph. F-Box Associated Domains = FBA1, FBA3, and FBD only. Other Domains = TUB, JmjC, LysM, WD40, zf_MYND, and DUF295. Only data from those experiments where the control *CCA1p::Luciferase* plants display a standard deviation less than 0.75 were included in our analyses.

DOI: https://doi.org/10.7554/eLife.44558.004

The following figure supplement is available for figure 2:

**Figure supplement 1.** RAE distributions of control plants in F-box experiments.

DOI: https://doi.org/10.7554/eLife.44558.005

the phase value of each individual T1 transgenic plant analyzed in the same experiment. Individual parental control plants were normalized in this same manner. Interestingly, we observed a phase shift in most of our decoy expressing populations when compared to the wild type, with the large majority showing a significant phase advance (*Figures 4–5*). This approximately 1 hr advance in phase (compare to *Figure 4—figure supplement 1* and *Figure 5—figure supplement 1*) appeared to be a general effect of transgene expression in our experiments, suggesting the phase of the *CCA1p::Luciferase* is particularly sensitive to transgene overexpression.

To overcome the general phase advance we compared each decoy population to the entire set of decoy populations for statistical testing. Using this method, we found that 40 F-box decoy populations cause a statistically significant change in phase (Welch's t-test with a Bonferroni corrected α of $1.94 \times 10^{-4}$), equating to approximately 22% of tested populations (*Figure 4*, marked with stars and blue gene names). We next set a cutoff of two hours phase difference to further subdivide our group into major (>2 hr) and minor (<2 hr) regulators. Based on this definition, many of the F-box decoy populations had minor phase differences (37/40 populations) while only three (*AT5G48980* – 2.36 hr delayed, *AT5G44980* – 2.16 hr delayed, and *AT5G42350* – 2.16 hr advanced) had major

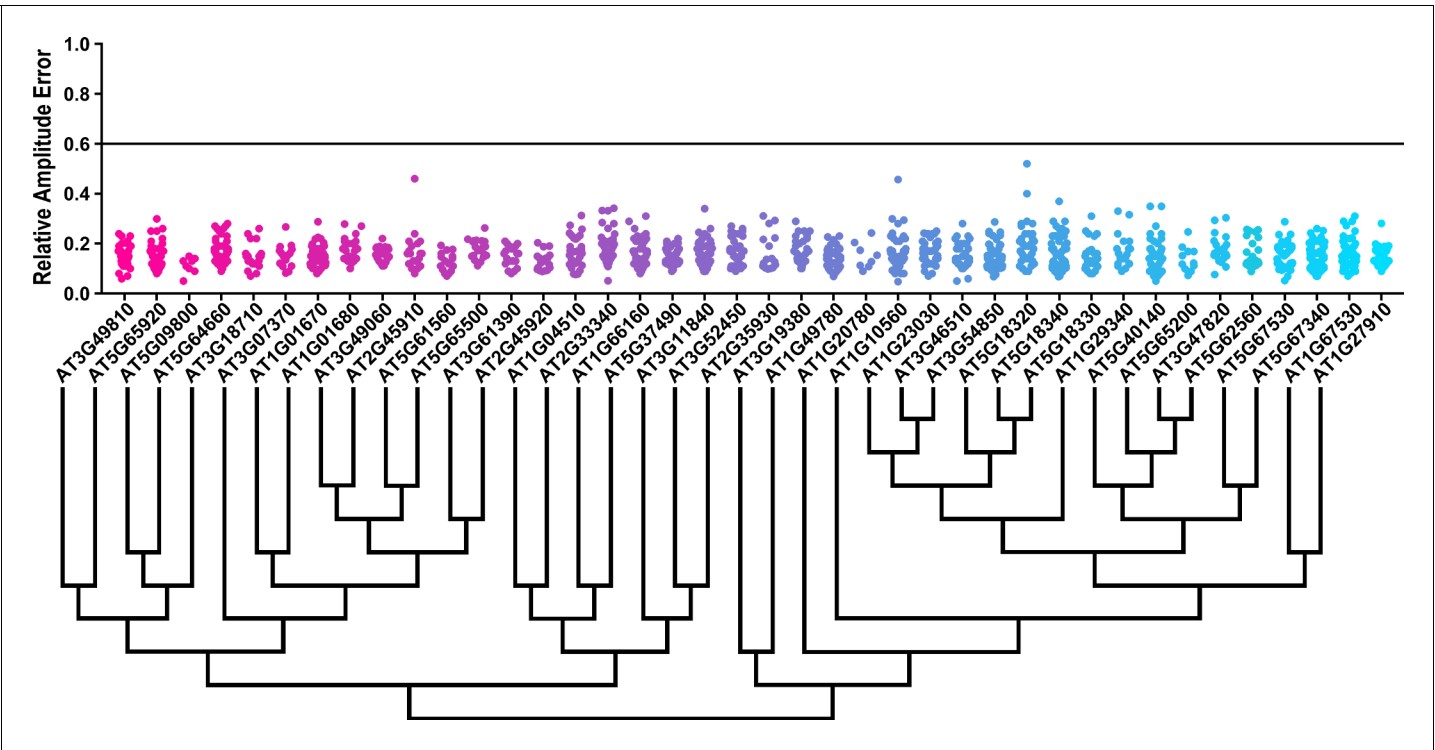

**Figure 3.** RAE distributions of U-box decoy plants. Values presented are the RAE for each individual T1 seedling. The black line represents the standard RAE cutoff of 0.6. Genes are ordered by closest protein homology using Phylogeny.Fr, (*Dereeper et al., 2008*), and a tree showing that homology is displayed beneath the graph.

DOI: https://doi.org/10.7554/eLife.44558.006

The following figure supplement is available for figure 3:

**Figure supplement 1.** RAE distributions of control plants in U-box experiments.

DOI: https://doi.org/10.7554/eLife.44558.007

phase differences. In addition to the F-box decoys, three of the U-box decoy populations have phase differences (Welch's t-test with a Bonferroni corrected α of $1.00 \times 10^{-3}$), none of which had major phase differences (*Figure 5*).

## Sequence and expression analysis of Phase-Regulating F-box proteins

Two of the three major phase regulators have not been studied previously. For this reason, we propose to name them *ALTERED CLOCK F-BOX* 1(*ACF1 – AT5G44980*) and *ACF2 (AT5G48980)*. As *AT5G42350* is already known as *COP9 INTERACTING F-BOX KELCH 1 (CFK1)*, we do not give this gene the *ACF* nomenclature.

In order to understand the function and regulation of the *ACFs*, we mined publically available expression data and the literature. ACF1, which contains both an F-box associated domain (FBD) and Leucine rich repeats (LRR), has no publications detailing its function. The absence of an identified phenotype may be due to the existence of a close homolog (*AT5G44960* – E-value of $1.60 \times 10^{-146}$, *Table 1*), as higher order mutants or dominant negative technology, such as the decoy technique, may be required to uncover its function.

Often genes that regulate clock function are also themselves regulated by the clock or light. Thus, we attempted to determine whether *ACF1* is regulated by the circadian clock or diurnal cycles. *ACF1* is not regulated by either light cycles or the circadian clock, as the correlation value, a measure of the similarity between the expression data and the hypothesized cycling pattern, is less than the standard correlation cutoff of 0.8 (*Mockler et al., 2007*). Furthermore, many core clock genes are expressed ubiquitously in the plant, although tissue specific clocks do exist (*Endo et al., 2014*; *Lee and Seo, 2018*; *Shimizu et al., 2015*). For this reason, we searched two expression atlases to determine the expression patterns of *ACF1*. Tissue expression maps suggest that *ACF1* is expressed

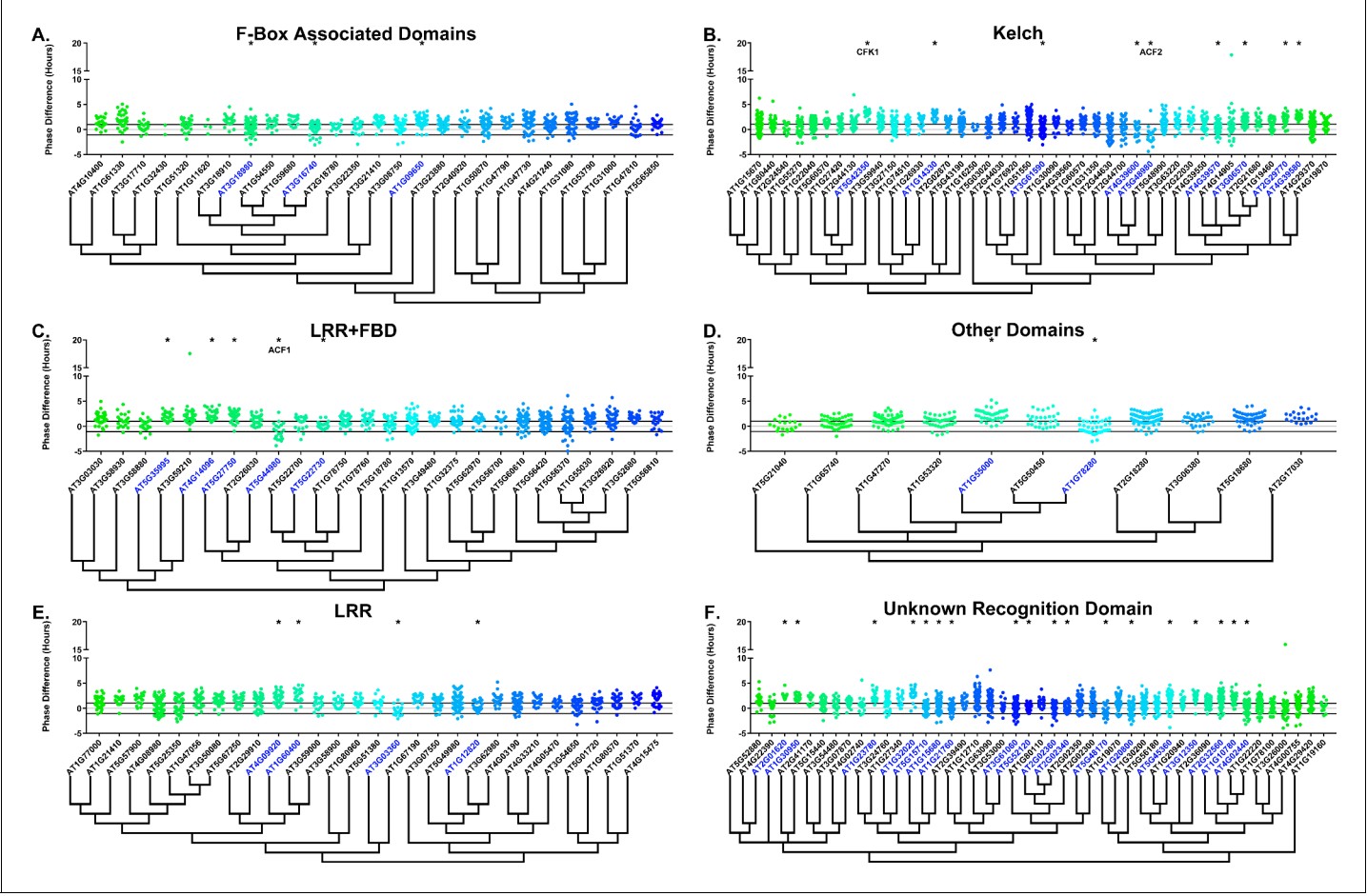

**Figure 4.** Phase distributions of F-box decoy plants. Values presented are the difference between the period of the individual decoy plant and the average period of the *CCA1p::Luciferase* control in the accompanying experiment. The grey line is at the average control value and the black lines are at ±the standard deviation of the control plants. Genes are separated by protein recognition domain and ordered by closest protein homology using Phylogeny.Fr, (*Dereeper et al., 2008*), and a tree showing that homology is displayed beneath the graph. F-Box Associated Domains = FBA1, FBA3, and FBD only. Other Domains = TUB, JmjC, LysM, WD40, zf_MYND, and DUF295. * and blue gene names = The entire population differs from wildtype with a Bonferroni-corrected $p<1.94\times10^{-4}$. Only data from those experiments where the control *pCCA1::Luciferase* plants display a standard deviation less than 0.75 were included in our analyses.

DOI: https://doi.org/10.7554/eLife.44558.008

The following figure supplement is available for figure 4:

**Figure supplement 1.** Phase distributions of control plants in F-box experiments.
DOI: https://doi.org/10.7554/eLife.44558.009

globally, although there may be some enrichment in senescent leaves or anthers (*Klepikova et al., 2016*; *Winter et al., 2007*). The global expression pattern suggests that *ACF1* has the potential to be involved in phasing of the circadian clock in all plant tissues.

We performed the same analysis on *ACF2*, which contains a Kelch repeat. No publications are available, possibly due to the existence of a close homolog (*AT5G48990* – E-value of $1.10 \times 10^{-124}$, *Table 1*), although *ACF2* was described in a manuscript discussing the prevalence of the Kelch-repeat containing F-box proteins in Arabidopsis (*Andrade et al., 2001*). Diurnal and circadian expression data was not available for this gene (*Mockler et al., 2007*). While data on *ACF2* expression is unavailable in one expression map, the second shows predominant expression in seed (*Klepikova et al., 2016*; *Winter et al., 2007*).

More is known about *CFK1*. While temporal expression data is unavailable, expression maps demonstrate that *CFK1* is expressed globally (*Table 1*) (*Klepikova et al., 2016*; *Mockler et al., 2007*; *Winter et al., 2007*), suggesting it is not tissue specific. *cfk1* knockout mutants cause

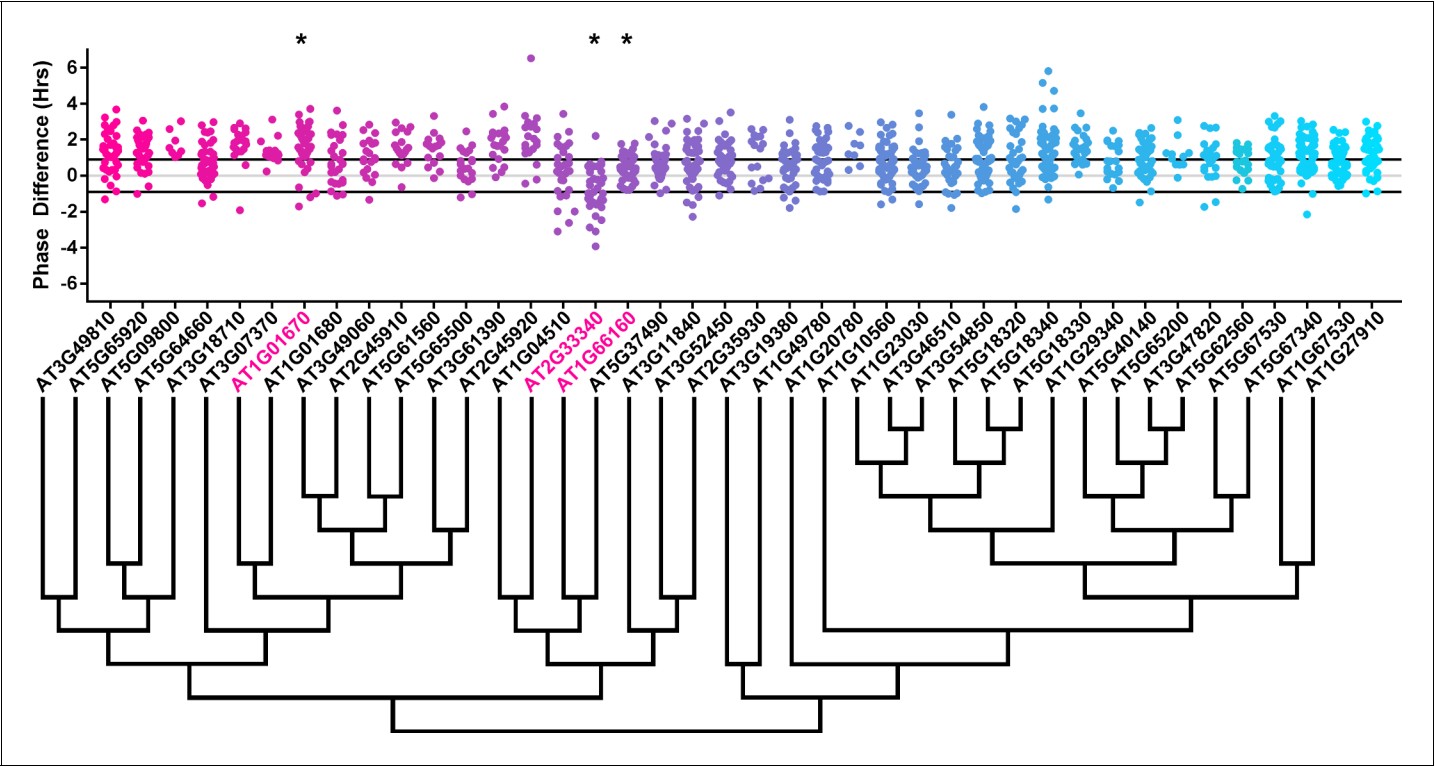

**Figure 5.** Phase distributions of U-box decoy plants. Values presented are the difference between the period of the individual decoy plant and the average period of the *CCA1p::Luciferase* control in the accompanying experiment. The grey line is at the average control value and the black lines are at ±the standard deviation of the control plants. Genes are ordered by closest protein homology using Phylogeny.Fr, (*Dereeper et al., 2008*), and a tree showing that homology is displayed beneath the graph. * and pink gene names = The entire population differs from wildtype with a Bonferroni-corrected p<1.00×10$^{-3}$.

DOI: https://doi.org/10.7554/eLife.44558.010

The following figure supplement is available for figure 5:

**Figure supplement 1.** Phase distributions of control plants in U-box experiments.
DOI: https://doi.org/10.7554/eLife.44558.011

decreased hypocotyl length, and interact with the CONSTITUTIVE PHOTOMORPHOGENIC 9 (COP9) Signalosome (*Franciosini et al., 2013*). *CFK1* has a very close homolog (*AT5G42360*, also known as *CFK2* – E-value of 3.6 × 10$^{-298}$), but while it cannot be completely redundant with *CFK1* because of the knockout phenotype, reduction in the levels of both genes increases the phenotypic severity (*Franciosini et al., 2013*). *CFK1* is expression induced by light (*Franciosini et al., 2013*), providing strength to the argument that it could be involved in clock function.

## The role of F-box decoys in clock period

Many clock E3 ubiquitin ligases control periodicity, thus we determined whether the decoys cause changes in clock period (*Godinho et al., 2007*; *Han et al., 2004*; *Liu et al., 2018*; *Reischl et al., 2007*). We calculated the period difference by calculating the average period of the parental control population in each experiment, then subtracting this value from the period value of each individual T1 transgenic plant analyzed in the same experiment. Unlike the effects on phase, we did not observe a general period shift across all decoy plants when doing this analysis (*Figure 6*, compare to *Figure 6—figure supplement 1*). This suggests that the period of the *pCCA1::Luciferase* reporter is not sensitive to general effects of transgene overexpression. From this analysis we found that 36 F-box decoy populations have significantly different periods than the control (Welch's t-test with a Bonferroni corrected α of 2.55 × 10$^{-4}$) (*Figure 6*, marked with stars and green gene names). These correspond to approximately 19% of tested populations (*Figure 6*, marked with stars and green gene names). We next divided the group into populations with minor (<1 hr period difference) and

**Table 1.** Publically available data for strong candidate F-box phase regulators.

Circadian expression data is from the Diurnal Project gene expression tool (*Mockler et al., 2007*). Tissue specific expression is from the Arabidopsis eFP browser (*Klepikova et al., 2016*; *Winter et al., 2007*). The closest homolog was determined by WU-BLAST2 using BLASTP and the Araport11 protein sequences database. N/A indicates that data is not available.

| Locus ID | Gene name | Circadian expression | | | | | | | | Tissue specific expression | | Closest homolog | | Publications (PMID) |
| --- | --- | --- | --- | --- | --- | --- | --- | --- | --- | --- | --- | --- | --- | --- |
| | | LDHC | | LL_LDHC | | LL12_LDHH | | LL23_LDHH | | Developmental map (eFP) | Klepikova atlas | Locus | E-Value | |
| | | Phase | Correlation | Phase | Correlation | Phase | Correlation | Phase | Correlation | | | | | |
| AT5G44980 | ACF1 | 16 | 0.50 | 23 | 0.38 | 14 | 0.69 | 4 | 0.43 | Global, especially senescent leaf | Global, stamen | AT5G44960 | $1.60 \times 10^{-146}$ | N/A |
| AT5G48980 | ACF2 | N/A | N/A | N/A | N/A | N/A | N/A | N/A | N/A | N/A | Silique with seed, seed | AT5G48990 | $1.10 \times 10^{-124}$ | N/A |
| AT5G42350 | CFK1 | N/A | N/A | N/A | N/A | N/A | N/A | N/A | N/A | Global, especially pollen | Global | AT5G42360 | $3.60 \times 10^{-298}$ | 23475998 |

DOI: https://doi.org/10.7554/eLife.44558.012

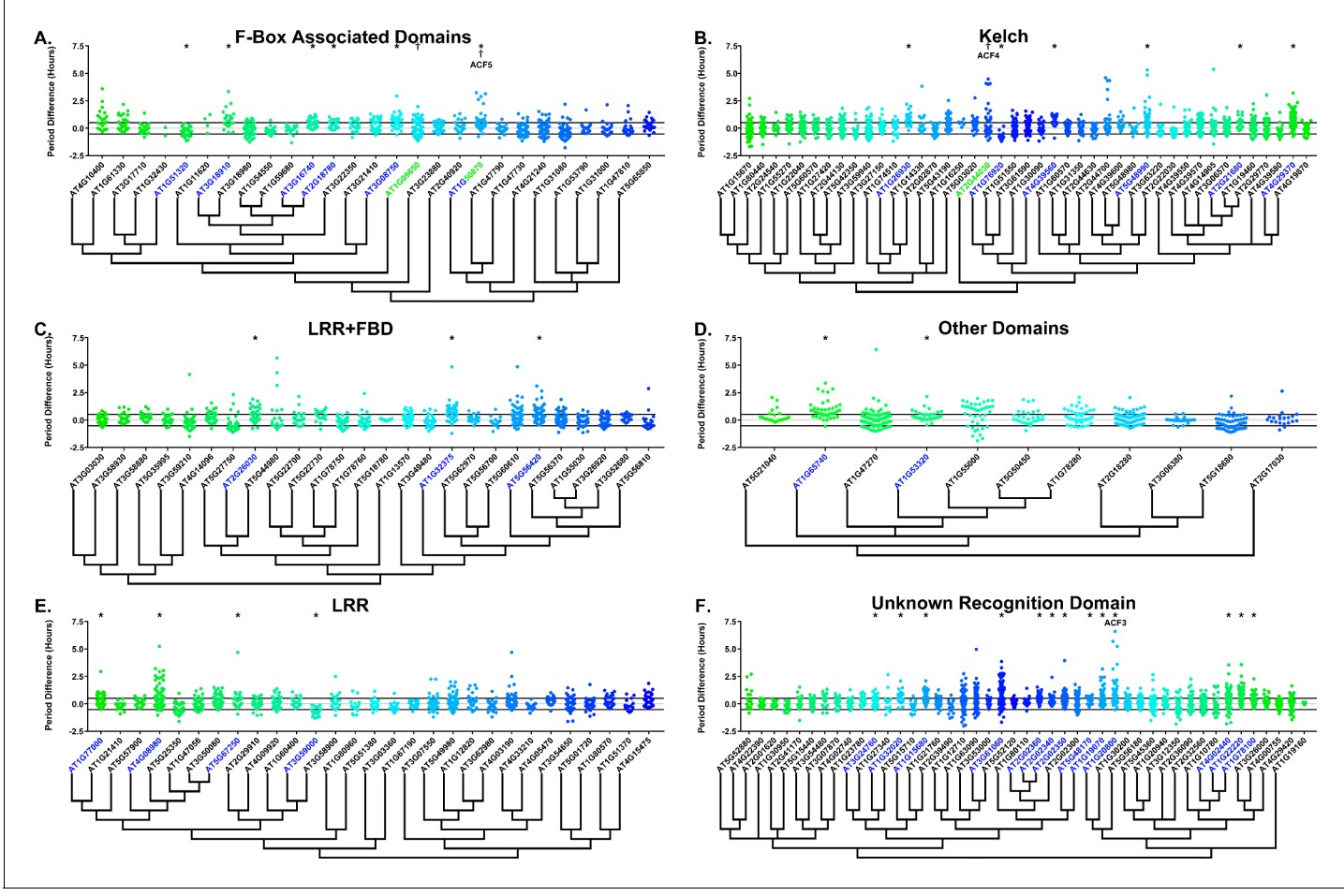

**Figure 6.** Period distributions of F-box decoy plants. Values presented are the difference between the period of the individual decoy plant and the average period of the *CCA1p::Luciferase* control in the accompanying experiment. The grey line is at the average control value and the black lines are at ±the standard deviation of the control plants. Genes are separated by protein recognition domain and ordered by closest protein homology using Phylogeny.Fr, (*Dereeper et al., 2008*), and a tree showing that homology is displayed beneath the graph. F-Box Associated Domains = FBA1, FBA3, and FBD only. Other Domains = TUB, JmjC, LysM, WD40, zf_MYND, and DUF295. * and blue gene names = The entire population differs from wildtype with a Bonferroni-corrected $p<2.55\times10^{-4}$; † and green gene names = A subset of the population differs from wildtype with a Bonferroni-corrected $p<2.55\times10^{-4}$. Only data from those experiments where the control *CCA1p::Luciferase* plants display a standard deviation less than 0.75 were included in our analyses.

DOI: https://doi.org/10.7554/eLife.44558.013

The following figure supplement is available for figure 6:

**Figure supplement 1.** Period distributions of control plants in F-box experiments.

DOI: https://doi.org/10.7554/eLife.44558.014

major (>1 hr period difference) effects on the period difference. Interestingly, only one F-box decoy population has a major period difference (*AT1G20800* – 1.02 hr longer than the wild type). The remaining have minor effects on the period difference.

Previously we showed that expressing decoys of clock-regulating F-box genes can result in separable subpopulations that affect circadian period differentially (*Lee et al., 2018*). We further analyzed the period data to identify decoy populations with statistically separable subpopulations. We define a subpopulation as a group of three or more decoy plants that have similar periods to each other but are statistically different than other subpopulations from the same decoy population (see the Materials and methods section for further details). There are two F-box decoy populations that are not different than the control as a whole population, but have distinct subpopulations that are different than the control (*Figure 6*, marked with daggers and blue gene names). One, *AT2G44030*, has a subpopulation that has a major effect on period, containing four plants with an average period

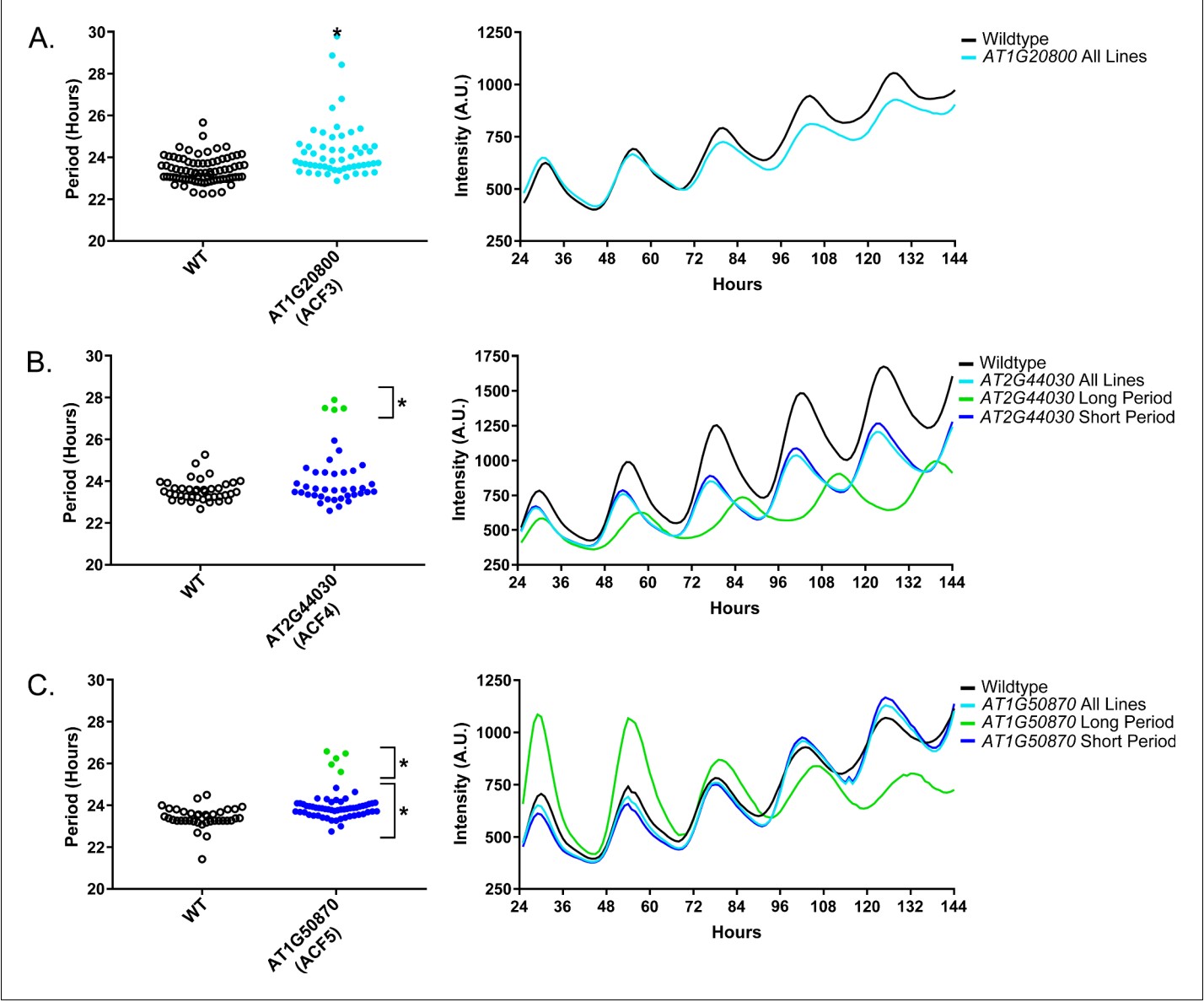

**Figure 7.** Circadian Phenotypes for selected high-priority F-box decoy populations. Period values and average traces for decoy plants with significant differences from the control across the entire population or a sub-population of plants greater than 1. Period values presented are raw period lengths as determined by *CCA1p::Luciferase*, and traces are calculated from the average image intensity across each seedling at each hour throughout the duration of the imaging experiment. Time 0 is defined as the dawn of the release into LL. A) *AT1G20800* (*ACF3*) decoy. B) *AT2G44030* (*ACF4*) decoy. C) *AT1G50870* (*ACF5*) decoy. Brackets define individual groups used for statistical testing against the wildtype control using a Welch's t-test with a Bonferroni-corrected α of $2.55 \times 10^{-4}$. * represents p<α. When multiple subpopulations were detected, the members of each group were separately averaged and presented in the traces along with the average of all plants.

DOI: https://doi.org/10.7554/eLife.44558.015

4.1 hr longer than the control. The remaining subpopulation is not significantly different from the control. A second decoy population, *AT1G50870*, which has a minor effect on the period overall (0.57 hr longer) also contains two separable subpopulations. One subpopulation has a major effect on the period (2.8 hr longer, n = 5), while the second has a minor effect (0.36 hr longer, n = 55).

Only three of the 41 identified populations or subpopulations had shorter periods than the wildtype. *AT1G76920* and *AT1G51320* both have shorter periods overall (0.76 and 0.29 hr shorter, respectively), while *AT1G09650* has a subpopulation that is 0.61 hr shorter than the wildtype. Nothing to date has been published regarding the functions of any of these genes. While none of these

**Table 2.** Publically available data for strong candidate F-box period regulators.
Circadian expression data is from the Diurnal Project gene expression tool (*Mockler et al., 2007*). Tissue specific expression is from the Arabidopsis eFP browser (*Klepikova et al., 2016*; *Winter et al., 2007*). The closest homolog was determined by WU-BLAST2 using BLASTP and the Araport11 protein sequences database. N/A indicates that data is not available.

| Locus ID | Gene name | Circadian expression | | | | | | | | Tissue specific expression | | Closest homolog | | Publications (PMID) |
| | | LDHC | | LL_LDHC | | LL12_LDHH | | LL23_LDHH | | Developmental map (eFP) | Klepikova atlas | Locus | E-Value | |
| | | Phase | Correlation | Phase | Correlation | Phase | Correlation | Phase | Correlation | | | | | |
| AT1G20800 | ACF3 | 18 | 0.51 | 4 | 0.66 | 15 | 0.65 | 16 | 0.66 | N/A | Young flower, stamen, ovule | AT1G20803 | $1.6 \times 10^{-87}$ | N/A |
| AT2G44030 | ACF4 | 18 | 0.73 | 5 | 0.54 | 6 | 0.62 | 22 | 0.79 | Seed, pollen | Young stamen, flower buds | AT3G46050 | $2.5 \times 10^{-59}$ | N/A |
| AT1G50870 | ACF5 | N/A | N/A | N/A | N/A | N/A | N/A | N/A | N/A | N/A | | AT1G47790 | $1.2 \times 10^{-102}$ | N/A |

DOI: https://doi.org/10.7554/eLife.44558.016

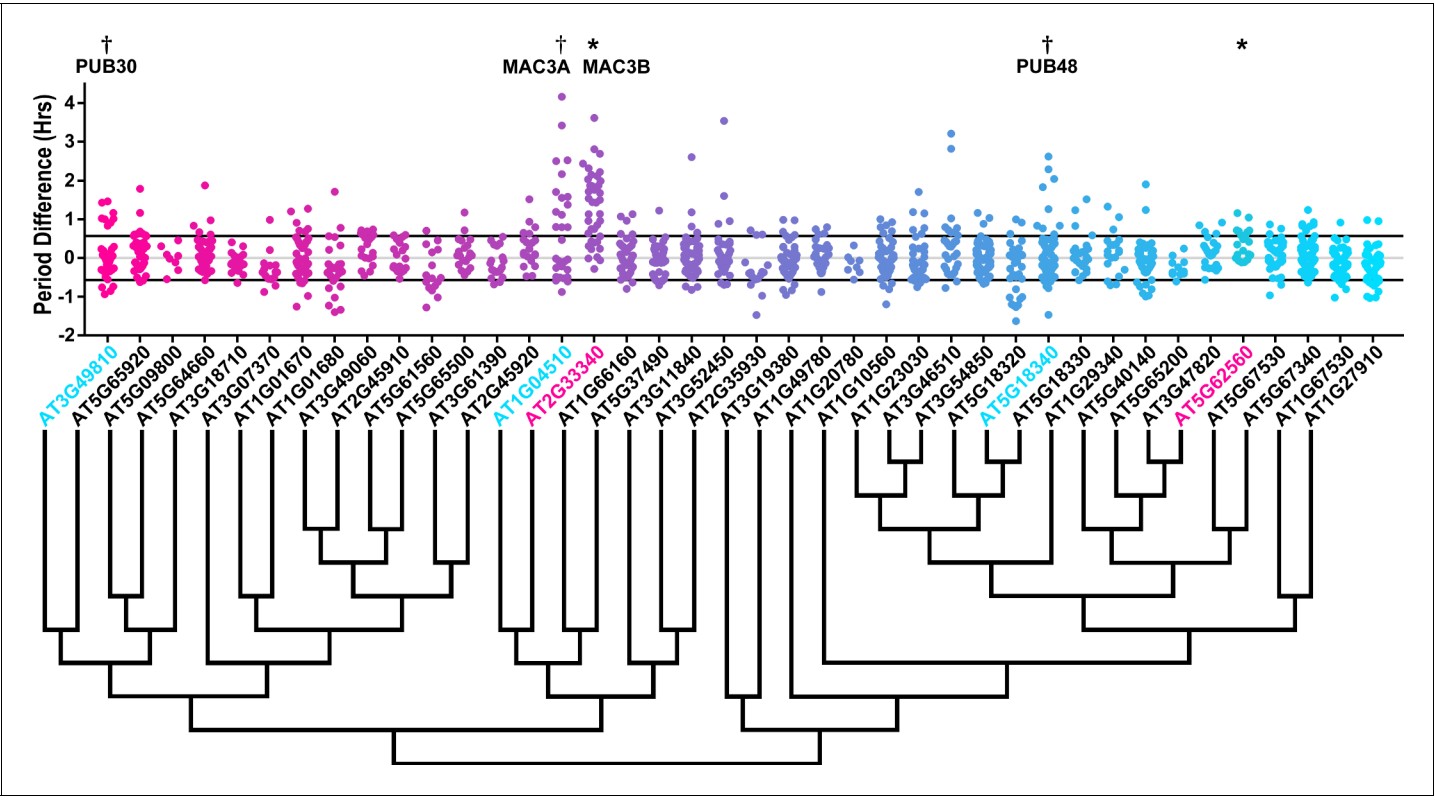

**Figure 8.** Period distributions of U-box decoy plants. Values presented are the difference between the period of the individual decoy plant and the average period of the *CCA1p::Luciferase* control in the accompanying experiment. The grey line is at the average control value and the black lines are at ±the standard deviation of the control plants. Genes are ordered by closest protein homology using Phylogeny.Fr, (*Dereeper et al., 2008*), and a tree showing that homology is displayed beneath the graph. * and pink gene names = The entire population differs from wildtype with a Bonferroni-corrected $p<1.09\times10^{-3}$; † and cyan gene names = A subset of the population differs from wildtype with a Bonferroni-corrected $p<1.09\times10^{-3}$.
DOI: https://doi.org/10.7554/eLife.44558.017

The following figure supplement is available for figure 8:

**Figure supplement 1.** Period distributions of control plants in U-box experiments.
DOI: https://doi.org/10.7554/eLife.44558.018

falls into the defined major effect category, the relative scarcity of short period effects make these potential candidates for follow-up study.

As a quality control measure we examine Luciferase reporter traces to identify any abnormalities in the rhythms of the decoy populations. We plotted the average Luciferase traces for the F-box decoy populations (or those with subpopulations) with major effects on period difference (*Figure 7*). Additionally, we have plotted the raw period data, color coded by subpopulation, so that the plants being included in each of the traces are obvious. The traces show that the decoys have relatively normal rhythms aside from the shifts in period, suggesting that the decoys are affecting period but not phasing and rhythmicity.

## Sequence and expression analysis of Period-Regulating F-box proteins

We also give the *ACF* nomenclature to the three genes with major period differences. *AT1G20800* we name *ACF3*, *AT2G44030* we name *ACF4*, and *AT1G50870* we name *ACF5*. We also performed the expression analyses and literature searches on these three *ACF* genes.

No publications exist regarding the functions of ACF3, which also has no known protein recognition domain. *ACF3* contains a close homolog (*AT1G20803* – E-value of $1.6\times10^{-87}$) which may be why it was not identified in previous forward genetic screens (*Table 2*). Expression of *ACF3* is not controlled by the circadian clock or diurnal cycles (*Mockler et al., 2007*). The two expression maps have widely varying expression data for *ACF3*, as one suggests that this gene is expressed globally,

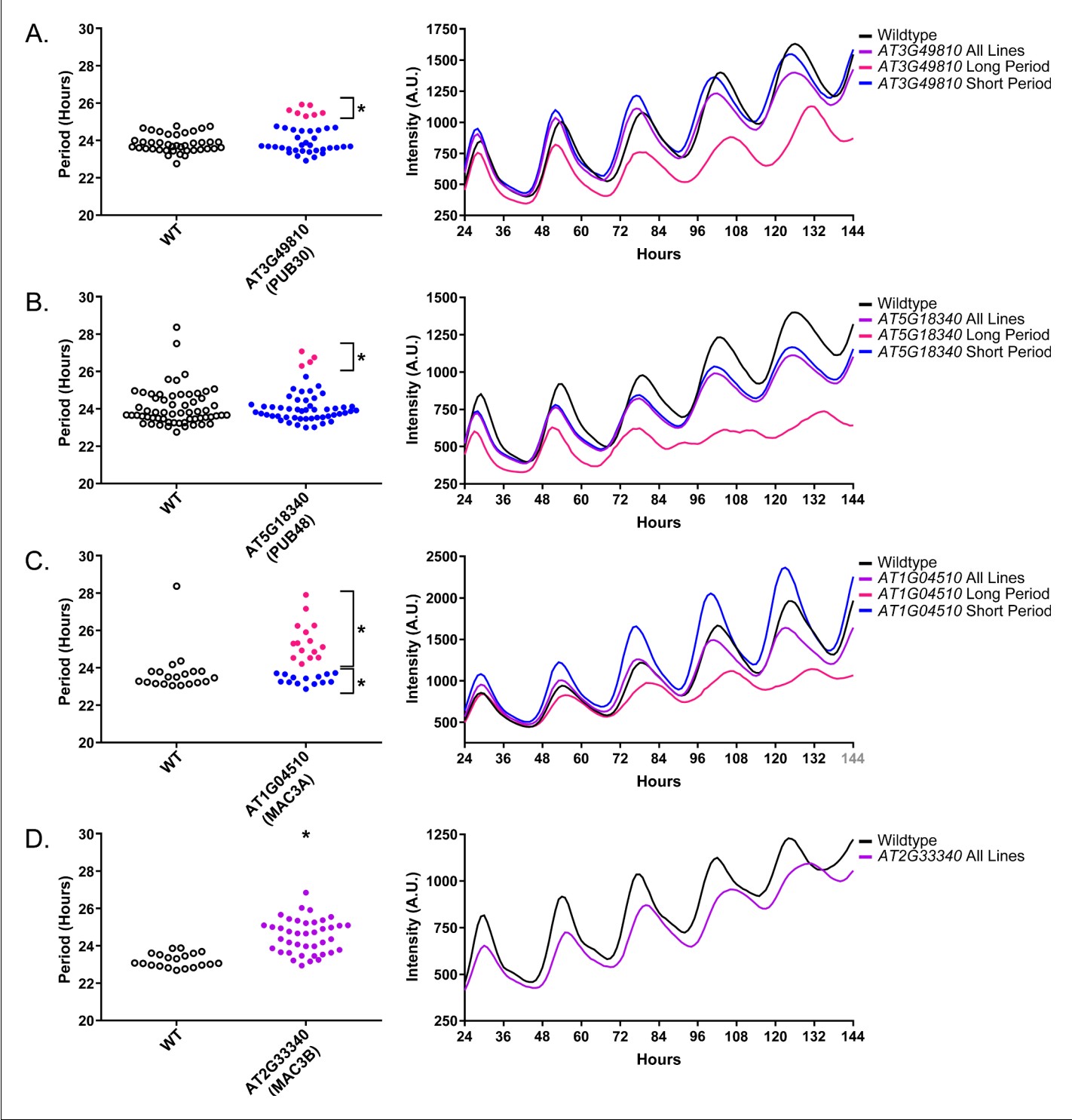

**Figure 9.** Circadian Phenotypes for selected high-priority U-box decoy populations. Period values and average traces for decoy plants with significant differences from the control across the entire population or a sub-population of plants greater than 1. Period values presented are raw period lengths as determined by *CCA1p::Luciferase*, and traces are calculated from the average image intensity across each seedling at each hour throughout the duration of the imaging experiment. Time 0 is defined as the dawn of the release into LL. (**A**) *AT3G49810* (*PUB30*) decoy. (**B**) *AT5G18340* (*PUB48*) decoy. (**C**) *AT104510* (*MAC3A*) decoy. D) *AT2G33340 (MAC3B)* decoy. Brackets define individual groups used for statistical testing against the wildtype control using a Welch's t-test with a Bonferroni-corrected α of $1.09 \times 10^{-3}$. * represents p<α. When multiple subpopulations were detected, the members of each group were separately averaged and presented in the traces along with the average of all plants.
DOI: https://doi.org/10.7554/eLife.44558.019

**Table 3.** Publically available data for strong candidate U-box period regulators.
Circadian expression data is from the Diurnal Project gene expression tool (**Mockler et al., 2007**). Tissue specific expression is from the Arabidopsis eFP browser (**Klepikova et al., 2016; Winter et al., 2007**). The closest homolog was determined by WU-BLAST2 using BLASTP and the Araport11 protein sequences database.

| Locus ID | Gene name | Circadian expression | | | | | | | | Tissue specific expression | | Closest homolog | | Publications (PMID) |
|---|---|---|---|---|---|---|---|---|---|---|---|---|---|---|
| | | LDHC | | LL_LDHC | | LL12_LDHH | | LL23_LDHH | | Developmental map (eFP) | Klepikova atlas | Locus | E-Value | |
| | | Phase | Correlation | Phase | Correlation | Phase | Correlation | Phase | Correlation | | | | | |
| AT5G18340 | PUB48 | 19 | 0.68 | 17 | 0.86 | 20 | 0.63 | 14 | 0.86 | Global, especially young flower and senescent leaf | Young flower bud, Stamen, mature leaf mature petiole | AT5G18320 | $3.8 \times 10^{-127}$ | 28077082 |
| AT1G04510 | MAC3A | 20 | 0.84 | 18 | 0.72 | 23 | 0.48 | 14 | 0.83 | Global | Global | AT2G33340 | $1.2 \times 10^{-233}$ | 29437988 28944490 19629177 |
| AT2G33340 | MAC3B | 15 | 0.49 | 8 | 0.56 | 17 | 0.80 | 8 | 0.73 | Global | Global | AT1G04510 | $7.2 \times 10^{-229}$ | 29437988 28944490 19629177 |
| AT3G49810 | PUB30 | 2 | 0.87 | 1 | 0.59 | 3 | 0.76 | 0 | 0.91 | Global | Everywhere except young seed and young silique | AT5G65920 | $1.2 \times 10^{-171}$ | 28865087 25410251 |

DOI: https://doi.org/10.7554/eLife.44558.020

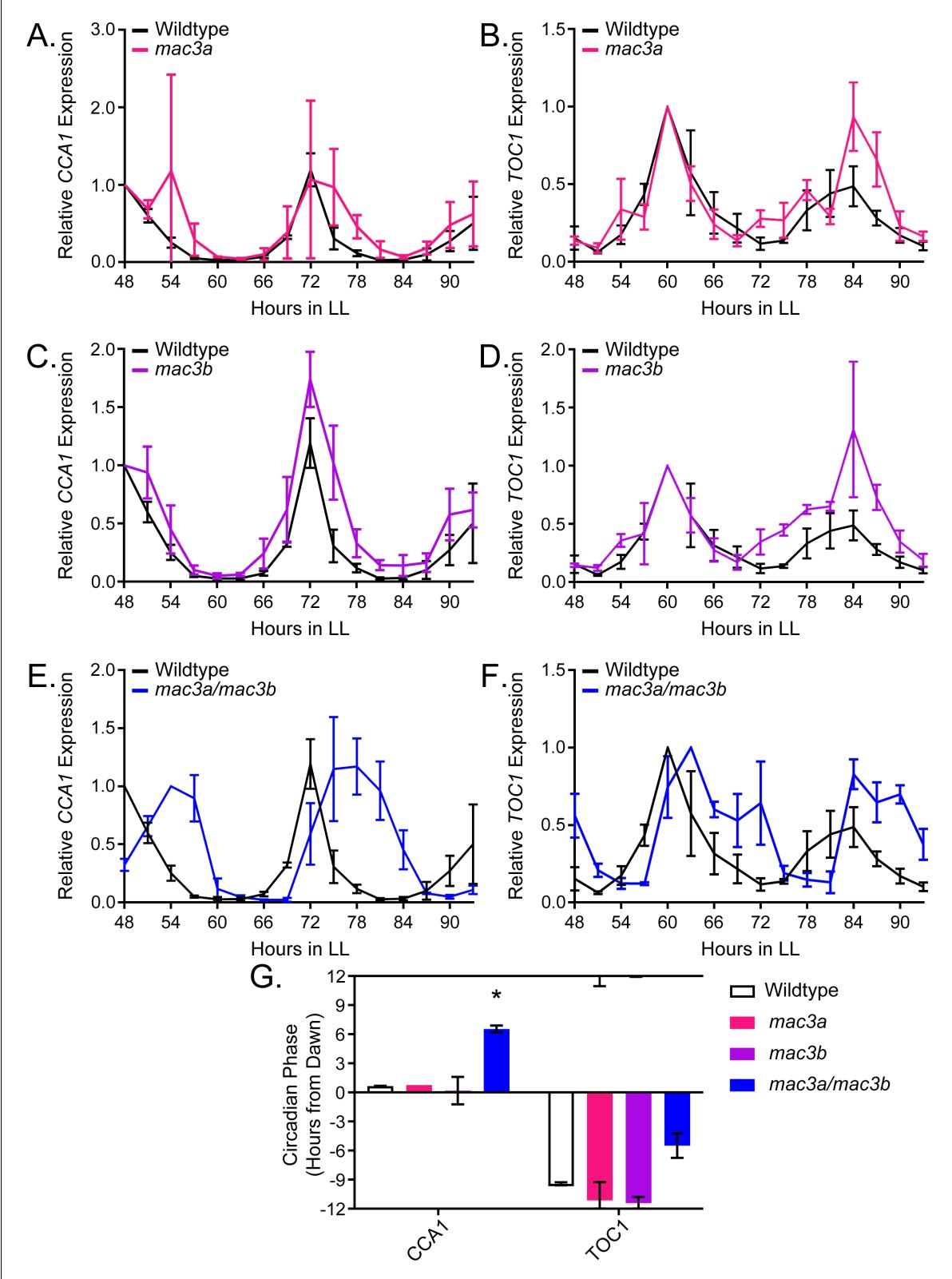

**Figure 10.** qRT-PCR of clock gene expression in *mac3a/mac3b* mutants. (A,C,E) *CCA1* and (B,D,F)*TOC1* expression was measured using quantitative RT-PCR in wildtype or homozygous (A-B) *mac3a*, (C-D) *mac3b*, and (E-F) *mac3a/mac3b* mutants under constant light conditions. Quantifications are the average of three biological replicates with error bars showing standard deviation. (G) FFT-NLLS analysis through the Biodare2 analysis platform shows
*Figure 10 continued on next page*

*Figure 10 continued*
altered phasing in the *mac3a/mac3b* double mutant. Error bars represent the standard deviation. * represents p<a Bonferroni-corrected alpha of 1.67 × 10$^{-2}$.
DOI: https://doi.org/10.7554/eLife.44558.021

while the other suggests that it is expressed exclusively in floral buds (*Klepikova et al., 2016*; *Winter et al., 2007*).

ACF4, which contains a Kelch-repeat domain, also has no described mutant phenotype, and is only described in an overview of the Kelch-repeat containing F-box proteins in Arabidopsis (*Andrade et al., 2001*). The closest homolog to *ACF4* (AT3G46050 – E-value of $2.5 \times 10^{-59}$) has not been studied (*Table 2*). As with *ACF3*, *ACF4* is not controlled by the circadian clock or light cycles (*Mockler et al., 2007*). *ACF4* is expressed globally in one developmental map and in floral buds in the other (*Klepikova et al., 2016*; *Winter et al., 2007*).

Very little is known about *ACF5* outside of its predicted FBA3 domain. No expression data, either diurnal or circadian regulation or tissue-specific, is available for *ACF5* (*Table 2*). Furthermore, no studies have been published on the function of this gene. It has some homology to other genes (AT1G47790 – E-value of $1.2 \times 10^{-102}$), which again indicates that it may have been missed in previous genetic screens due to redundancy.

## The role of U-box decoys in clock period

We also analyzed the U-box library data to identify decoy populations or subpopulations with period differences (*Figure 8* and *Figure 8—figure supplement 1*). We found that two populations of decoy plants had average period lengths longer than wild type, AT2G33340 (1.31 hr longer) and AT5G62560 (0.37 hr longer) (Welch's t-test with a Bonferroni corrected $\alpha$ of $1.09 \times 10^{-3}$) (*Figure 8*, marked with stars and pink gene names) making *AT2G33340* the only U-box decoy with a major effect on the average period difference.

Three additional U-box decoy populations had subpopulations that were different from the parental control, *AT3G49810*, *PUB48*, and *AT1G04510* (*Figure 8*, marked with daggers and blue gene names). In this case all three had subpopulations that we consider strong regulators. *AT3G49810* has one subpopulation which is 1.11 hr longer than the wildtype (n = 7), while the other subpopulation is not significantly different (n = 32). Similarly, *PUB48* has one subpopulation which is 2.20 hr longer than the wildtype (n = 4), and a second subpopulation which is not significantly different (n = 48). *AT1G04510* is different in that it has one subpopulation that we consider in the major effect category (1.75 hr longer, n = 15), and one that is short period (0.4 hr shorter, n = 14). Because these subpopulations alter the period differently, the overall average is not statistically significant from wildtype (*Figure 8*).

Again we examined Luciferase reporter traces to identify any abnormalities in the rhythms of the decoy populations. We plotted the average Luciferase traces for the U-box decoy populations (or those with subpopulations) with major effects on period difference (*Figure 9*). Additionally, we have plotted the raw period data, color coded by subpopulation, so that the plants being included in the traces are obvious. *AT3G49810*, *AT1G04510*, and *AT2G33340* decoy populations or subpopulations have period defects but otherwise normal rhythms. *AT5G18340* long period plants, on the other hand, show some rhythmic abnormality. The traces appear to decrease in rhythmicity on hour 84 (*Figure 9b*) but then regain rhythmicity by hour 120. Interestingly, some of the control plants for this experiment have abnormally long periods suggesting some stochastic noise in this experiment (*Figure 9b*). Although this subpopulation and the controls pass our stringent statistical filters, the trace and raw period data may suggest that the results need to be interpreted carefully for *AT5G18340* and further experimentation will be required to confirm its role in the clock.

## Sequence and expression analysis of Period-Regulating U-box proteins

As the U-box genes have been given their PUB nomenclature (*Azevedo et al., 2001*; *Yee and Goring, 2009*), we did not rename these genes. We do, however, perform the same expression and literature searches that we performed on the *ACF* genes.

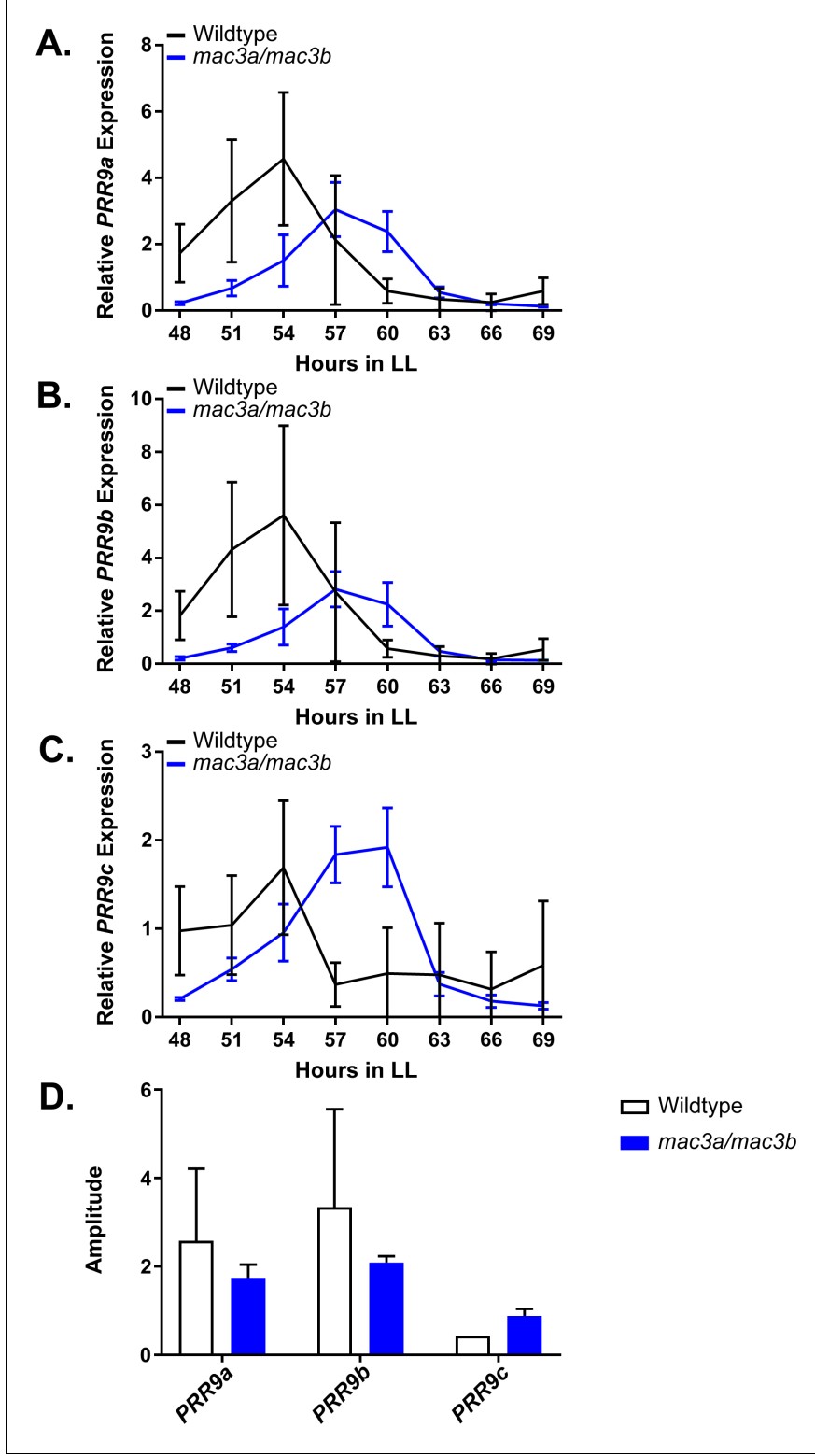

**Figure 11.** qRT-PCR of PRR9 splicing in *mac3a/mac3b* mutants. *PRR9* isoform expression was measured using quantitative RT-PCR in *mac3a/mac3b* mutants. (A) *PRR9a*, (B) *PRR9b*, and (C) *PRR9c* isoforms were analyzed. Quantifications are the average of three biological replicates with error bars showing standard deviation. (D) FFT-NLLS analysis through the Biodare2 platform suggests altered isoform amplitude in the *mac3a/mac3b* double mutant. Error bars represent the standard deviation.

DOI: https://doi.org/10.7554/eLife.44558.022

*AT3G49810*, also known as *PUB30*, contains two armadillo (ARM) repeats and has a potential homolog, *PUB31* (*AT5G65920*; E-value of $1.2 \times 10^{-171}$, *Table 3*). *PUB30* expression is rhythmic under diurnal conditions and under one of the circadian conditions (*Mockler et al., 2007*). Both tissue expression maps suggest that *PUB30* is expressed globally (*Klepikova et al., 2016*; *Winter et al., 2007*). *PUB30* has a described function in inhibiting the salt stress response (*Hwang et al., 2015*; *Zhang et al., 2017*).

*AT5G18340* is known as *PUB48*, and, like *PUB30*, contains two ARM repeats and has a partially redundant homolog, *PUB46* (*AT5G18320*, E-value of $3.8 \times 10^{-127}$, *Table 3*) (*Adler et al., 2017*). *PUB48* expression is not rhythmic under diurnal cycles. Interestingly, it is rhythmic under circadian conditions in two out of three available experiments, and has a similar phase in both (ZT14 and ZT17). Expression profiling suggests that *PUB48* is expressed globally, although it exhibits some enrichment in floral buds and potentially senescent leaves (*Klepikova et al., 2016*; *Winter et al., 2007*). *PUB48* is also involved in stress responses, as it has been shown to positively regulate the response to drought stress (*Adler et al., 2017*).

*AT1G04510* and *AT2G33340* are also known as *MAC3A* and *MAC3B*, respectively. *MAC3A* and *MAC3B* both contain 7 WD repeats, are 82% identical at the protein sequence level, and are known to act redundantly (*Table 3*) (*Monaghan et al., 2009*). While both cycle under one of the circadian conditions, only *MAC3A* cycles under diurnal conditions (*Mockler et al., 2007*). Tissue-specific expression profiles are very similar for the two genes, as both are expressed globally (*Klepikova et al., 2016*; *Winter et al., 2007*). *MAC3A* and *MAC3B* are orthologous to the human and yeast Pre-mRNA Processing factor 19 (PRP19) proteins, the central components of the spliceosome activation machinery known as the Nineteen Complex (NTC) (*Chanarat and Sträßer, 2013*; *Hogg et al., 2010*). In plants, they were initially identified regulators of plant immunity, but further work confirmed their roles in the regulation of splicing and miRNA biogenesis (*Jia et al., 2017*; *Li et al., 2018*; *Monaghan et al., 2009*). Accordingly, splicing is a critical regulatory step in plant clock function (*Filichkin and Mockler, 2012*; *Park et al., 2012*; *Sanchez et al., 2010*; *Simpson et al., 2016*; *Wang et al., 2012*). Because the *MAC3A* and *MAC3B* decoys affect period, we hypothesize that they are redundantly controlling clock function through regulated splicing of clock genes.

## MAC3A and MAC3B are functionally redundant U-box proteins that regulate plant circadian clock function

To prove the validity of our decoy screening platform we performed detailed genetic and molecular follow-up experiments on *MAC3A* and *MAC3B*, two potentially redundant regulators of the plant circadian clock. We grew the *mac3a* and *mac3b* single knockout mutants (*Monaghan et al., 2009*) in LD conditions for 10 days and transferred them to constant light for two days. We collected tissue from the plants every three hours for two days and performed qRT-PCR to measure expression of the core clock genes, *CCA1* and *TOC1*. The *mac3a* and *mac3b* single mutants alone have little effect on the period, amplitude, or phase of the circadian clock (*Figure 10A–D*) suggesting their functions may be redundant. Thus, we obtained the *mac3a/mac3b* double mutant and monitored *CCA1* and *TOC1* expression under the same conditions as the single mutants (*Monaghan et al., 2009*). Unlike the single mutants, the *mac3a/mac3b* double mutant has a significant phase delay (*Figure 10E–F*). We quantified the phases using FFT-NLLS on the Biodare2 platform (biodare2.ed.ac.uk *Zielinski et al., 2014*). The *mac3a/mac3b* double mutant has a phase delay of 3.95 hr for *TOC1* expression and 5.90 hr for *CCA1* expression (*Figure 10G*). While the phase delay in *CCA1* expression is statistically significant, we do not reach statistical significance for the *TOC1* phase delay. This is likely due to the appearance of a secondary peak in *TOC1* expression in the *mac3a/mac3b* double mutant that makes phase calls difficult (*Figure 10F*). The appearance of a secondary peak in the *TOC1* expression profile suggests a more complex alteration of the circadian clock in the *mac3a/mac3b* double mutant than a simple phase delay. However, the phase delay in *CCA1* is consistent with the effects caused by a lengthened clock period, similar to the period lengthening observed in the *MAC3A* and *MAC3B* decoy populations. Together, this suggests that *MAC3A* and *MAC3B* are *bona fide* regulators of the circadian clock, and the first U-box genes, to our knowledge, identified as regulators of the circadian clock in any system. Furthermore, the absence of a clock phenotype in the *mac3a* and *mac3b* single mutants demonstrates the genes are redundant, and highlights the strength of the decoy technique to overcome traditional genetic barriers.

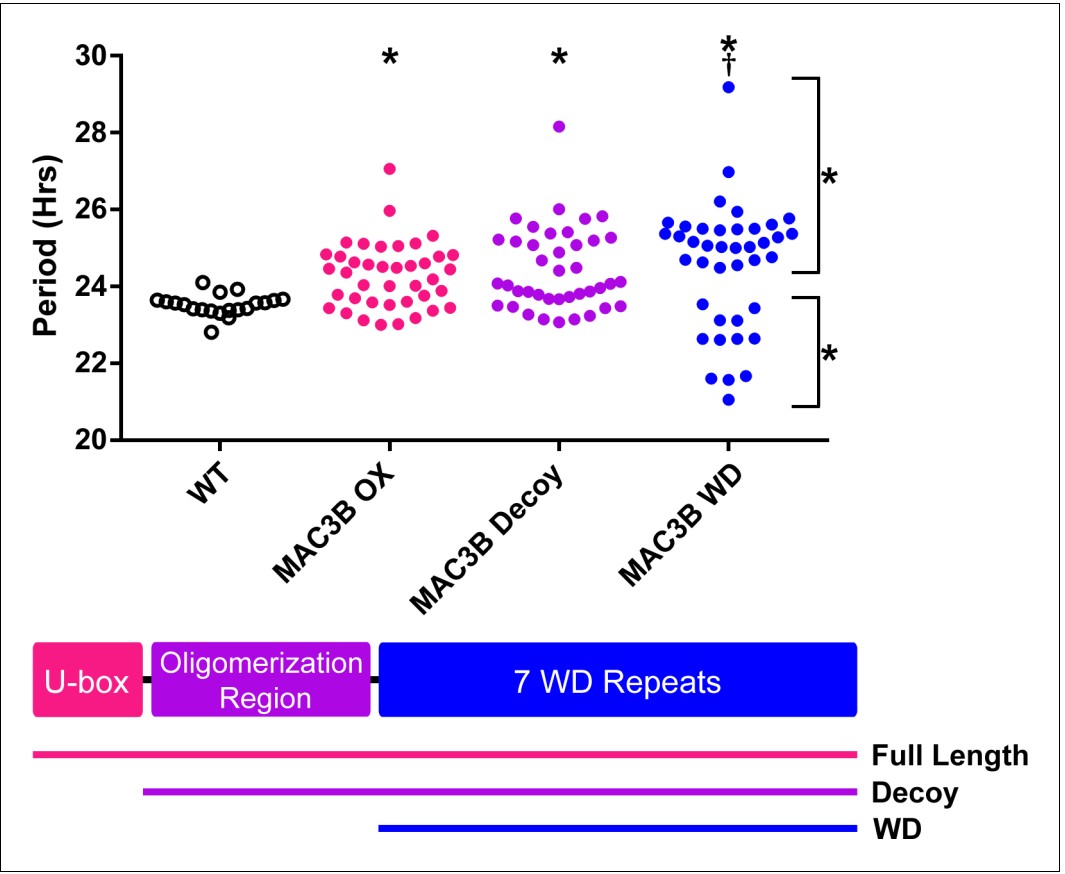

**Figure 12.** Period analyses of MAC3B overexpression constructs. Period was measured in T1 *MAC3B* full length, *MAC3B* decoy, and *MAC3B* WD insertion plants. Period values presented are raw period values measured by *CCA1p::Luciferase* expression. A schematic of which domains are in each construct is included below.
DOI: https://doi.org/10.7554/eLife.44558.023

## MAC3A and MAC3B control splicing of a plant circadian clock gene

As the *mac3a/mac3b* mutant was recently shown to exhibit global splicing defects and intron retention, we hypothesized that these splicing defects may impact the circadian clock. Like *MAC3A* and *MAC3B*, another component of the NTC, *SNW/SKI-interacting protein* (*SKIP*), has a lengthened circadian period when mutated, likely due in part to the dysregulation of *PRR9* and *PRR7* splicing

**Table 4.** Selected IP-MS results from the MAC3B decoy.
MAC3B decoy peptide hits are from one IP-MS experiment using the MAC3B decoy as the bait. Combined control peptide hits are summed from the independent control experiments of wildtype Col-0 and *35S::His-FLAG-GFP* expressing plants.

| Locus | Protein name | Total spectral counts | |
|---|---|---|---|
| | | **MAC3B decoy** | **Combined controls** |
| AT2G33340 | MAC3B | 294 | 15 |
| AT1G04510 | MAC3A | 129 | 0 |
| AT1G09770 | CDC5 | 36 | 0 |
| AT1G07360 | MAC5A | 24 | 44 |
| AT3G18165 | MOS4 | 35 | 0 |
| AT4G15900 | MAC2/PRL1 | 19 | 23 |

DOI: https://doi.org/10.7554/eLife.44558.024

(*Wang et al., 2012*). For these reasons, we performed time course qRT-PCR on the *mac3a/mac3b* mutant to investigate *PRR9* splicing. In the *mac3a/mac3b* mutant we observe a decrease in the amplitude of the active *PRR9* spliceoform, termed *PRR9a*, as well a secondary inactive form, referred to as *PRR9b* (*Figure 11A–B*). This is accompanied by an increase in the average amplitude of the inactive spliceoform, *PRR9c*, which retains an intron inappropriately (*Figure 11C*). We quantified these differences in amplitude using the Biodare2 analysis platform (biodare2.ed.ac.uk; *Zielinski et al., 2014*), and observed a similar trend, although the large degree of variability in the wildtype does not allow the data to reach statistical significance (*Figure 11D*). These results are consistent with previous results showing that mutations in splicing factors result in elevated *PRR9c* expression. Together, this data suggests that *MAC3A* and *MAC3B* play a role in the circadian clock by promoting the proper splicing of clock components.

## Perturbations in *MAC3B* expression lead to circadian clock defects

Accurate expression of genes involved in the circadian clock is essential to maintaining 24 hr periodicity (*Más et al., 2003*; *Rawat et al., 2011*; *Somers et al., 2004*). For this reason, we tested the effects of constitutive expression of full-length *MAC3B* on circadian clock function. We created transgenic plants expressing FLAG-His-MAC3B under the control of a 35S constitutive promoter in the *CCA1p::Luciferase* reporter background. Interestingly, constitutive expression of the full-length *MAC3B* causes period lengthening similar to *mac3a/mac3b* double mutant (*Figure 12*). This indicates that maintaining proper expression of *MAC3B* is necessary for clock function.

## MAC3B decoys form biologically relevant complexes

We have previously shown that F-box decoy proteins are able to interact with target proteins and regulatory partners and retain the ability to form biologically relevant complexes (*Lee et al., 2018*). We tested whether the U-box decoy proteins are similarly capable of interacting with the same proteins as the full length U-box proteins. The decoy proteins contain a 3XFLAG-6XHis affinity tag for immunoprecipitation, thus we performed an immunoprecipitation experiment with the MAC3B decoy and analyzed interacting proteins via mass spectrometry (IP-MS). As control we performed immunoprecipitation with a 3XFLAG-6XHis tagged GFP transgenic line. From the list of potential interacting proteins (*Supplementary file 2*) we identified known components of the NTC complex. We compared this list to the previously identified components of the plant NTC, and identified five common components in addition to the MAC3B bait peptides, three of which (MAC3A, CELL DIVISION CYCLE 5 (CDC5), and MODIFIER OF SNC1,4 (MOS4)) were only found in the MAC3B IP-MS experiments and not in the controls (*Table 4*) (*Monaghan et al., 2009*). This data suggests that the MAC3B decoy is capable of forming biologically relevant complexes in vivo and importantly supports the idea that the decoy strategy can be used for genetic and biochemical analyses of E3 ligase function in plants.

We observed MAC3A and MAC3B interacting in our IP/MS data and PRP19, the MAC3A and MAC3B orthologue from yeast and humans, is predicted to form a tetramer (*Grillari et al., 2005*; *Ohi et al., 2005*). Tetramer formation is predicted to be mediated by a conserved coiled coil that is present in MAC3A and MAC3B (*Li et al., 2018*), and was included in our MAC3A and MAC3B decoy constructs (*Figure 12*). We wanted to test the importance of the dimerization domain in clock function using our decoy system. Thus, we created a MAC3B decoy construct which expresses the WD repeats without the predicted oligomerization domain (MAC3B WD). Constitutive expression of the MAC3B WD domain causes two subpopulations of transgenic plants, one with lengthened period (25.5 hr) and one with shortened period (22.5 hr) (*Figure 12*). This was different than expressing the decoy or full-length MAC3B and indicates that oligomerization plays an important role in MAC3B function in the clock. Interestingly, the MAC3B WD phenotype more closely resembles the MAC3A decoy phenotype, suggesting that MAC3A and MAC3B may have diverged slightly in their biochemical function or oligomerization capabilities.

## Discussion

### Summary

To overcome genetic redundancy we performed a large-scale reverse genetic screen of plant E3 ubiquitin ligases. We generated transgenic plants expressing dominant-negative E3 ubiquitin ligase decoys and determined the effects on the circadian clock. From this screen we identified the first U-box-type E3 ubiquitin ligases, MAC3A and MAC3B, involved in circadian clock function along with additional putative 'major' and 'minor' clock period and phase regulators. Importantly, our detailed follow-up studies show that *MAC3A* and *MAC3B* are redundant regulators of clock function making it unlikely that they would be identified using traditional forward genetic screening methods. These genes represent the second family of E3 ubiquitin ligases to be identified in circadian clock function in plants, and their discovery uncovers a connection between three large cellular networks: the circadian clock, the ubiquitin proteasome, and splicing.

### MAC3A and MAC3B are part of the NTC and affect clock splicing

Splicing is a critical regulatory step in plant circadian clock function, and mutation of another component of the NTC, *SKIP*, lengthens circadian similar to the *mac3a/mac3b* double mutant. The lengthened period in the *skip* mutant is likely due to the dysregulation of splicing of *PRR9*, *PRR7*, and other circadian genes (*Wang et al., 2012*). In concordance, we show that the *mac3a/mac3b* double mutant has defects in *PRR9* splicing, but it is also likely that other clock genes may have spliceoform imbalances in the *mac3a/mac3b* mutant as well (*Jia et al., 2017*).

*MAC3A* and *MAC3B* were identified as interacting partners of MOS4, a positive regulator of plant innate immunity. Genetic experiments then demonstrated that they are also required for plant immunity (*Monaghan et al., 2009*). They were named MOS4-Associated-Complex 3A and B (MAC3A and MAC3B) but share homology with the spliceosome activating component PRP19. In both yeast and humans, PRP19 is known to be the core component of a large complex, known as the NTC. The NTC plays key roles in DNA repair, cell cycle progression and genome maintenance, and activating the spliceosome (*Chanarat and Sträßer, 2013*; *Hogg et al., 2010*). Further work has since shown that *MAC3A* and *MAC3B* (alternatively *PUB59* and *PUB60* or *PRP19a* and *PRP19b*) are involved in the global regulation of splicing in plants (*Jia et al., 2017*). These results, along with the identification of plant NTC components interacting with MAC3A and MAC3B in this study and others, strongly suggest that these genes are core components of the plant NTC (*Monaghan et al., 2009*).

Because the clock is an interlocked series of feedback loops it can be difficult to predict how mutation or misexpression of a gene will affect clock function. Interestingly, this is the same for *MAC3B* in which constitutive expression of *MAC3B* results in a similar period defect as the *mac3a/mac3b* double mutant. There are two possible explanations: 1) MAC3A and MAC3B are involved in splicing various clock genes and the cumulative effect of their disruption is a lengthened period, or 2) splicing of clock genes is both positively and negatively regulated and disruption of this balance causes lengthened period. It is possible to distinguish between these two events by determining the full extent of clock gene mis-splicing in the *mac3a/mac3b* mutant and *MAC3B* overexpression plants, and then performing complementation experiments with the cDNA of the mis-spliced clock genes. Evidence from the *MAC3A* decoy and *MAC3B* WD populations indicates that it may be the former, as we observed short period plants in both populations. The absence of short period plants in the *MAC3B* decoy is interesting, and suggests that these two proteins may have diverged slightly in their functions. Domain swap experiments or conversion of key residues between MAC3A and MAC3B may further elucidate these differences.

### The decoy technique uncovers difficult-to-identify clock regulators by overcoming redundancy

Our genetic studies of *MAC3A* and *MAC3B* highlight a critical strength of the decoy technique as a screening platform, the ability to overcome genetic redundancy. We observe minimal effect on the circadian clock in single *mac3a* and *mac3b* mutants. Yet, the double mutant has a strong non-additive genetic effect, suggesting the *MAC3A* and *MAC3B* genes are redundant. Exhaustive traditional forward genetic screens for clock mutants have not identified *MAC3A* and *MAC3B* (*Hazen et al.,*

*2005*; *Jones et al., 2012*; *Kevei et al., 2006*; *Martin-Tryon et al., 2007*; *Somers et al., 2000*), sug-gesting that reverse genetic strategies, such as our decoy approach, are important for generating a comprehensive genetic picture of the plant circadian clock.

## Description of candidate clock genes

In addition to *MAC3A* and *MAC3B*, we also identified 43 F-box and U-box genes which are likely to be involved in the regulation of the phase circadian clock, and 42 genes which are likely to be involved in the regulation of the period. We have highlighted seven of these as high priority regulators due to their strong phenotypic effects. These genes include *ACF1* (*AT5G44980*), *ACF2* (*AT5G48980*), *CFK1*, *ACF3* (*AT1G20800*), *ACF4* (*AT2G44030*), *ACF5* (*AT1G50870*), *PUB48*, and *PUB30*.

Nearly nothing is known about four of the high-priority *ACF*s that we discovered in our study. While future work will be required to untangle the relationships between these three genes and the circadian clock, our ability to isolate these novel genes highlights the strengths of the decoy library. All three genes have close homologs (*Table 2*), so it is possible that, similar to *MAC3A* and *MAC3B*, forward genetic screens failed to identify these genes due to redundancy.

*CFK1* is a regulator of hypocotyl length, and *CFK1* expression is light-induced (*Franciosini et al., 2013*). Light is known to control both the phasing of the circadian clock as well as hypocotyl elongation, so the ability of *CFK1* to respond to light signals makes it a promising candidate. Furthermore, *CFK1* is believed to be a target of the COP9 signalosome (CSN). The CSN is required for proper rhythmicity in *Neurospora*, and CSN mutants lead to impaired phase resetting in *Drosophila* (*He et al., 2005*; *Knowles et al., 2009*; *Zhou et al., 2012*). It is possible that the CSN plays a similar role in plants, possibly through regulation of *CFK1*. Future work detailing these connections between *CFK1*, the CSN, and the plant circadian clock would likely be quite fruitful.

In our U-box screen functions for putative hits are partially understood. *PUB48* and *PUB30* are involved in stress responses. *PUB30* is involved in the response to high salt conditions, and mutants exhibit reduced salt tolerance (*Hwang et al., 2015*; *Zhang et al., 2017*). Perturbations in the circadian clock can lead to altered salt stress response, so it is possible that alterations in the circadian clock of the *pub30* mutants similar to what we observe with the *PUB30* decoy leads to the observed salt stress phenotype (*Kim et al., 2013*; *Nakamichi et al., 2009*). *PUB48* is involved in response to drought stress (*Adler et al., 2017*). Although drought tolerance has not been implicated as an input to the circadian clock, water intake has been demonstrated to be a clock output (*Takase et al., 2011*). It is possible that altering the circadian clock in these plants may lead to improper regulation of water intake, and thus cause the drought stress sensitivity phenotype. Alternatively, recent work demonstrates that humidity cycles are sufficient to entrain the plant circadian clock, and contributes to rhythmicity even under cycling light conditions (*Mwimba et al., 2018*). *PUB48* may play a role in regulating this input pathway, and thus lead to both altered drought sensitivity and an altered circadian clock. Future work on investigating the intersection between PUB48, water levels, and the circadian clock should help illuminate these connections.

All of the candidate regulators were identified using our reverse genetic decoy approach. It is imperative that further genetic and molecular work be performed to confirm their roles in clock function. Full-length overexpression or mutant studies would be informative; however, as demonstrated by our work on *MAC3A* and *MAC3B*, higher order mutants may be needed in order to uncover the functions of redundant genes. It is our hope that the candidate genes identified in this manuscript will serve as a springboard for future work in the field.

## Usage of the decoy library

Here we have described the creation of the decoy library and its application to the identification of circadian clock regulators. However, the decoy technique and library is broadly applicable to other areas of plant biology. We will provide the library in four formats: *E. coli* containing the pENTR vectors for each decoy; *E. coli* containing the decoy recombined in the 35S::HIS-FLAG vector; *A. tumefaciens* containing the decoy recombined in the 35S::HIS-FLAG vector; and pooled populations of T2 transgenic Arabidopsis Col-0 seeds expressing the HIS-FLAG decoy. Each of these resources can be used in various ways.

*The pENTR library in E. coli*: The pENTR library allows any individual to recombine the decoy E3 ubiquitin ligases into any destination vector of interest. This can be done for any individual gene or *en masse* for the purposes of screening. This would allow one to drive the expression of the decoy E3 ligase under a variety of other promoters, including tissue-specific, temporally-specific, inducible, or low expression promoters. Additionally, the decoy library could be particularly potent for doing protein-protein interaction studies or studies of protein stability. One advantage of the decoys is that they prevent false negatives that can arise in protein-interaction studies involving E3 ligases due to substrate degradation in heterologous systems. Some potential uses include yeast-two-hybrid, mammalian cell culture expression, or insect cell culture expression. Furthermore, the library could be transformed into vectors for expression in transient plant systems. These uses are numerous but include bimolecular fluorescence complementation (BiFC), proximity labeling, fluorescence resonance energy transfer (FRET). Additionally, transient protein stability assays could be performed including *in planta* co-expression with potential targets to look for increases in stability or co-expression in heterologous mammalian expression systems (*Lee et al., 2018*). There are likely a large number of other uses for this library.

*The 35S::HIS-FLAG decoy library in E. coli and A. tumefaciens:* Our study centered on transformation of a clock reporter line, demonstrating the utility of this library for identifying genetic regulators in non-traditional genetic backgrounds. The *A. tumefaciens* library could be used similarly in almost any genetic or marker transgenic background. We have provided this library with both kanamycin and glufosinate resistance to facilitate transformation even in transgenic backgrounds with either resistance. We can imagine the potency of this for both enhancer and suppressor reverse genetic screens. This library could also be transformed into lines containing tagged proteins to identify E3 ligases that control their stability. One example would be transforming the decoy library into a transgenic line expressing a putative proteasomal target protein fused to GFP. T1 transgenics could be examined on selection plates for increased fluorescence to identify the E3 ligase that has an effect on the protein stability. Additionally, transient protein degradation assays could be performed in a similar way by co-expressing the decoys with a putative proteasomal target fused to any measurable marker. Furthermore, transient expression followed by physiological assays or measurement of any molecular marker could identify the E3 ubiquitin ligases that are important for a variety of cellular processes. This library could also be transformed into additional Arabidopsis ecotypes or other species that are transformed by agrobacterium.

*T2 transgenic Col-0 seeds:* Our results suggest that the analysis of a single homozygous transformant line is often misleading. In our screening process we analyzed 20 T1 plants per decoy population and were able to identify phenotypic subpopulations. Optimally, a similar strategy screening a group of 20 or more T1 plants should allow users to identify E3 ligases involved in their biological process of interest. However, we recognize that the labor that is required to retransform a library of this size can limit the usability. For this reason, the library will be provided as pools of T2 transgenics that originated from approximately 20 individual T1 transgenic plants. These can be used for any physiological, cellular, chemical, or conditional genetic screen. We can imagine many creative ways this library could be employed by the scientific community.

## Conclusions

The data presented in this manuscript demonstrates that decoys are a potent and scalable technique for identifying the function of plant E3 ubiquitin ligases. The molecular and genetic reagents generated in the course of this study are already available to the wider scientific community. We have shown here that this technique applies to multiple families of E3 ubiquitin ligases, the U-boxes and the F-boxes. It is a logical extension to believe this technique would work for other families of E3 ubiquitin ligases, so long as the domains involved in interaction with the E2 conjugating enzyme are easily defined. We believe that the data shown here demonstrates that the decoy technique is a valuable resource to anyone interested in uncovering the function of plant E3 ubiquitin ligases involved in any aspect of plant biology.

## Materials and methods

### Key resources table

| Reagent type (species) or resource | Designation | Source or reference | Identifiers | Additional information |
|---|---|---|---|---|
| Genetic Reagent (*Arabidopsis thaliana*) | *CCA1p::Luciferase* | **Pruneda-Paz et al., 2009** | | Dr. Pruneda-Paz |
| Genetic Reagent (*A. thaliana*) | *mac3a* | **Monaghan et al., 2009** | | Dr. Xin Li |
| Genetic Reagent (*A. thaliana*) | *mac3b* | Arabidopsis Biological Resource Center | SALK 144856C | |
| Genetic reagent (*A. thaliana*) | *mac3a/mac3b* | **Monaghan et al., 2009** | | Dr. Xin Li |
| Antibody | Monoclonal mouse anti-FLAG antibody | Sigma | Cat#: F3165 RRID:AB_259529 | Conjugated to Dynabeads M-270 Epoxy (Thermo Fisher Scientific, cat# 14311D) |

## Construction of decoy libraries

In order to create F-box and U-box decoys, the CDS annotation from TAIR10 was compared to the protein domain annotation from Uniprot, and the nucleotide boundaries for the start and end of the E3 ubiquitin ligase domain were recorded. As the majority of F-box proteins follow a standardized domain architecture with the F-box domain located in the N-terminus of the protein, decoy constructs were created by removing the nucleotides of the F-box and any nucleotides upstream of the F-box domain. The U-box proteins, however, do not follow a standardized domain architecture, as the U-box domain can be located anywhere throughout the protein. For this reason, for the purposes of cloning the decoy library, U-boxes were sorted into three categories: N-terminal U-boxes, C-terminal U-boxes, and central U-boxes. Those genes where the U-box domain began less than 75 amino acids from the start of the protein were sorted into the N-terminal U-box class; those genes where the U-box domain ends less than 75 amino acids from the end of the protein were sorted into the C-terminal U-box class; and those genes where the U-box domain begins more than 75 amino acids from the beginning of the protein and ends more than 75 amino acids from the end of the protein were sorted into the central U-box class. 75 amino acids was selected as the threshold as there were gaps in U-box distribution throughout the protein sequence which made this a natural choice.

Primers for creation of F-box and U-box decoys were designed using the CDS annotation from TAIR10 (**Supplementary file 3**). For central U-boxes, N-terminal and C-terminal constructs were generated using PCR products generated from cDNA, then overlap extension PCR was used to fuse N-terminal and C-terminal constructs into the full decoy construct. PCR products generated from cDNA were inserted into pENTR/D-TOPO vectors (Invitrogen, cat. # K240020) then transferred into pB7-HFN, pK7-HFN, and pB7-HFC destination vectors using LR recombination (**Huang et al., 2016a**). F-boxes and N-terminal U-boxes were cloned into pB7-HFN and pK7-HFN (N-terminal His-FLAG tags), C-terminal U-boxes were cloned into pB7-HFC (C-terminal HIS-FLAG tags), and central U-boxes were cloned into all three vectors. The decoy constructs were transformed into Arabidopsis Col-0 expressing the circadian reporter *pCCA1::Luciferase* or Col-0 by the floral dip method using *Agrobacterium tumefaciens* GV3101 (**Pruneda-Paz et al., 2009**).

## Phenotypic screening

Control *pCCA1::Luciferase* and decoy seeds were surface sterilized in 70% ethanol and 0.01% Triton X-100 for 20 min prior to being sown on ½ MS plates (2.15 g/L Murashige and Skoog medium, pH 5.7, Cassion Laboratories, cat#MSP01% and 0.8% bacteriological agar, AmericanBio cat# AB01185) with or without appropriate antibiotics (15 μg/ml ammonium glufosinate (Santa Cruz Biotechnology, cat# 77182-82-2) for vectors pB7-HFN and pB7-HFC, or 50 μg/ml kanamycin sulfate (AmericanBio) for pK7-HFN). Seeds were stratified for two days at 4°C, then transferred to 12 hr light/12 hr dark conditions for seven days. Twenty seven-day old seedlings from each genotype were arrayed on 100 mm square ½ MS plates in a 10 × 10 grid, then treated with 5 mM D-luciferin (Cayman Chemical Company, cat# 115144-35-9) dissolved in 0.01% TritonX-100. Seedlings were imaged at 22°C under constant white light provided by two LED light panels (Heliospectra L1) with light fluence rate of 100

$\mu$mol m$^{-2}$ s$^{-1}$. The imaging regime is as follows: each hour lights are turned off for two minutes, then an image is collected using a five minute exposure on an Andor iKon-M CCD camera; lights remain off for one minute after the exposure is completed, then lights return to the normal lighting regime. The CCD camera was controlled using Micromanager, using the following settings: binning of 2, pre-am gain of 2, and a 0.05 MHz readout mode (*Edelstein et al., 2014*). Using this setup, 400 seedlings are simultaneously imaged across four plates. Images are acquired each hour for approximately six and a half days. Data collected between the first dawn of constant light and the dawn of the sixth day are used for analyses.

The mean intensity of each seedling at each time point was calculated using ImageJ (*Schneider et al., 2012*). The calculated values were imported into the Biological Rhythms Analysis Software System (BRASS) for analysis. The Fast Fourier Transform Non-linear Least Squares (FFT-NLLS) algorithm was used to calculate the period, phase, and relative amplitude from each individual seedling (*Moore et al., 2014*).

## Data normalization and statistical analysis

To allow for comparison across independent imaging experiments, period and phase data was normalized to the individual wildtype control performed concurrently. The average value of the wildtype control plants was calculated for every experiment, then this average was subtracted from the value of each individual T1 insertion or control wildtype plant done concurrently. This normalized value was used for statistical analyses.

The presence of sub-populations was determined by a custom MATLAB script which takes the normalized values as inputs and creates histograms of each population. The peaks and troughs of the histogram are identified, and number of seedlings within each peak (between each pair of troughs) was counted. Peaks were discarded if the number of seedlings was too small (less than 3), if there was only one bin between peaks, or if the difference between peak and trough was too small (less than 3). If the number of remaining peaks was two or more, the population was defined as having subpopulations. The locations of the troughs in the histogram were used as the division point to sort plants into their respective subpopulations. The source code for this custom MATLAB script is available as a supplemental file.

Welch's t-test was used to compare each normalized T1 insertion plant population or subpopulation to the population of normalized control plants. For period, all wildtype plants were used as the control. For phase, the entire population of decoy plants was used as the control. Data from the F-box decoy library was treated as an independent experiment from data from the U-box decoy library. In order to decrease the number of false positives caused by multiple testing, we utilized a Bonferroni corrected $\alpha$ as the p-value threshold. The $\alpha$ applied differs between experiments, and is noted throughout.

## Measurement of circadian gene expression in *mac3a/mac3b* mutants

Homozygous *mac3a/mac3b* mutant plants in the Col-0 background were generated previously (*Monaghan et al., 2009*). Single and double *mac3a* and *mac3b* mutant and Col-0 seeds were grown on ½ MS plates and entrained in 12 hr light/12 hr dark conditions at a fluence rate of 130 $\mu$mol m$^{-2}$ s$^{-1}$ at 22°C. 10 day old seedlings were transferred into constant light conditions for 48 hr prior to the start of the time course. Seedlings were collected in triplicate every three hours for two days starting at ZT0 and snap-frozen using liquid nitrogen, then ground using the Mixer Mill MM400 system (Retsch). Total RNA was extracted from ground seedlings using the RNeasy Plant Mini Kit and treated with RNase-Free DNase (Qiagen, cat#74904 and 79254) following the manufacturer's protocols. cDNA was prepared from 1 $\mu$g total RNA using iScript Reverse Transcription Supermix (Bio-Rad, cat#1708841), then diluted 15-fold and used directly as the template for quantitative real-time RT-PCR (qRT-PCR). The qRT-PCR was performed using 3.5 $\mu$l of diluted cDNA and 5.5 $\mu$M primers listed in *Supplementary file 3* (*Czechowski et al., 2004*; *Farré et al., 2005*; *Lee and Thomashow, 2012*) using iTaq Universal SYBR Green Supermix (Bio-Rad, cat# 1725121) with the CFX 384 Touch Real-Time PCR Detection System (Bio-RAD). The qRT-PCR began with a denaturation step of 95°C for 3 min, followed by 45 cycles of denaturation at 95°C for 15 s, and primer annealing at 53°C for 15 s. Relative expression of *CCA1* and *TOC1* was determined by the comparative $C_T$ method using *IPP2* (*AT3G02780*) as an internal control. The relative expression levels represent the mean values of

$2^{-\Delta\Delta CT}$ from three biological replicates, where $\Delta CT = C_T$ of the decoy – $C_T$ IPP2 and the reference point is the first peak time for each replicate (ZT0 for Col-0, *mac3a*, and *mac3b*, and ZT6 for *mac3a/mac3b* for *CCA1* expression, and ZT12 for Col-0, *mac3a*, and *mac3b*, and ZT15 for *mac3a/60* for *TOC1* expression).

## Measurement of PRR9 spliceoforms

*mac3a/mac3b* double mutant seedlings in the Col-0 background and parental Col-0 seeds were grown and harvested as described for circadian gene expression analysis. qPCR was performed as described for *CCA1* expression analysis. Primers used in a previous study (*Wang et al., 2012*) to track *PRR9* spliceoform expression are shown in *Supplementary file 3*. The relative expression levels represent the mean values of $2^{-\Delta\Delta CT}$ from three biological replicates, where $\Delta CT = C_T$ of the decoy – $C_T$ IPP2 and the reference point is ZT0 from one of the Col-0 replicates.

## Immunoprecipitation and mass spectrometry of MAC3B decoys

Individual T1 *pB7-HFN-MAC3B* transgenic plants in a Col-0 background and control Col-0 and *pB7-HFC-GFP* were grown as described for phenotype analysis. Seven-day old seedlings were transferred to soil and grown under 16 hr light/8 hr dark at 22°C for 2–3 weeks. Prior to harvest, plants were entrained to 12 hr light/12 hr dark at 22°C for 1 week. Approximately 40 mature leaves from each background was collected and flash frozen in liquid nitrogen, such that each sample was a mixture of leaves from multiple individuals to reduce the effects of expression level fluctuations. Tissue samples were ground in liquid nitrogen using the Mixer Mill MM400 system (Retsch). Immunoprecipitation was performed as described previously (*Huang et al., 2016a*; *Huang et al., 2016b*; *Lu et al., 2010*). Briefly, protein from 2 ml tissue powder was extracted in SII buffer (100 mM sodium phosphate pH 8.0, 150 mM NaCl, 5 mM EDTA, 0.1% Triton X-100) with cOmplete EDTA-free Protease Inhibitor Cocktail (Roche, cat# 11873580001), 1 mM phenylmethylsμlfonyl fluoride (PMSF), and PhosSTOP tablet (Roche, cat# 04906845001) by sonication. Monoclonal mouse anti-FLAG antibodies (Sigma cat# F3165) were cross-linked to Dynabeads M-270 Epoxy (Thermo Fisher Scientific, cat# 14311D) for immunoprecipitation. Immunoprecipitation was performed by incubation of protein extracts with beads for 1 hr at 4°C on a rocker. Beads were washed with SII buffer three times, then twice in F2H buffer (100 mM sodium phosphate pH 8.0, 150 mM NaCl, 0.1% Triton X-100). Beads were eluted twice at 4°C and twice at 30°C in F2H buffer with 100 μg/ml FLAG peptide, then incubated with TALON magnetic beads (Clontech, cat# 35636) for 20 min at 4°C, then washed twice in F2H buffer and three times in 25 mM Ammonium Bicarbonate. Samples were subjected to trypsin digestion (0.5 μg, Promega, cat# V5113) at 37°C overnight, then vacuum dried using a SpeedVac before being dissolved in 5% formic acid/0.1% trifluoroacetic acid (TFA). Protein concentration was determined by nanodrop measurement (A260/A280)(Thermo Scientific Nanodrop 2000 UV-Vis Spectrophotometer). An aliquot of each sample was further diluted with 0.1% TFA to 0.1 μg/μl and 0.5 μg was injected for LC-MS/MS analysis at the Keck MS and Proteomics Resource Laboratory at Yale University.

LC-MS/MS analysis was performed on a Thermo Scientific Orbitrap Elite mass spectrometer equipped with a Waters nanoAcquity UPLC system utilizing a binary solvent system (Buffer A: 0.1% formic acid; Buffer B: 0.1% formic acid in acetonitrile). Trapping was performed at 5 μl/min, 97% Buffer A for 3 min using a Waters Symmetry C18 180 μm x 20 mm trap column. Peptides were separated using an ACQUITY UPLC PST (BEH) C18 nanoACQUITY Column 1.7 μm, 75 μm x 250 mm (37°C) and eluted at 300 nl/min with the following gradient: 3% buffer B at initial conditions; 5% B at 3 min; 35% B at 140 min; 50% B at 155 min; 85% B at 160–165 min; then returned to initial conditions at 166 min. MS were acquired in the Orbitrap in profile mode over the 300–1,700 m/z range using one microscan, 30,000 resolution, AGC target of 1E6, and a full max ion time of 50 ms. Up to 15 MS/MS were collected per MS scan using collision induced dissociation (CID) on species with an intensity threshold of 5000 and charge states two and above. Data dependent MS/MS were acquired in centroid mode in the ion trap using one microscan, AGC target of 2E4, full max IT of 100 ms, 2.0 m/z isolation window, and normalized collision energy of 35. Dynamic exclusion was enabled with a repeat count of 1, repeat duration of 30 s, exclusion list size of 500, and exclusion duration of 60 s.

The MS/MS spectra were searched by the Keck MS and Proteomics Resource Laboratory at Yale University using MASCOT (*Perkins et al., 1999*). Data was searched against the SwissProt_2015_11. fasta *Arabidopsis thaliana* database with oxidation set as a variable modification. The peptide mass tolerance was set to 10 ppm, the fragment mass tolerance to 0.5 Da, and the maximum number of allowable missed cleavages was set to 2.

## Resource distribution

The F-box and U-box decoy library is available from the Arabidopsis Biological Resource Center (ABRC) (http://abrc.osu.edu) as pENTR-decoy, pB7-HFN-decoy, pK7-HFN-decoy, and pB7-HFC-decoy as appropriate. Additionally a collection of T2 seeds generated from a mixed population of T1s expressing an individual pB7-HFN-decoy, pK7-HFN-decoy, or pB7-HFC-decoy construct is also available from ABRC. pB7-HFN-decoy, pK7-HFN-decoy, and pB7-HFC-decoy are available as *A. tumefaciens* stocks directly from the authors upon request. ABRC stock numbers for the available seed stocks and constructs can be found in *Supplementary file 4*.

# Acknowledgements

We would like to thank Christopher Adamchek, Cathy Chamberlin, Suyuna Eng Ren, Sandra Pariseau, Denise George, Brandon Williams, Milan Sandhu, and Annie Jin for their technical support. We would also like to thank the Keck Proteomics Facility at Yale for their assistance with proteomics, and Dr. Xin Li for providing the *mac3a* and *mac3a/mac3b* double mutants. Additionally, we would like to thank Dr. Qingqing Wang and Dr. Bryan Thines for their helpful comments on the manuscript. This work was supported by the National Science Foundation (EAGER #1548538) and the National Institutes of Health (R35 GM128670) to JMG; by a Rudolph J Anderson Fund Fellowship to C-ML; by a Forest BH and Elizabeth DW Brown Fund Fellowship to C-ML and WL; by the National Institutes of Health (T32 GM007499), the Gruber Foundation, and the National Science Foundation (GRFP DGE-1122492) to AF.

# Additional information

## Funding

| Funder | Grant reference number | Author |
|---|---|---|
| National Science Foundation | EAGER #1548538 | Joshua M Gendron |
| National Institutes of Health | R35 GM128670 | Joshua M Gendron |
| Gruber Foundation | | Ann Feke |
| Rudolph J. Anderson Fund | | Chin-Mei Lee |
| Forest B.H. and Elizabeth D.W. Brown Fund | | Wei Liu |
| National Science Foundation | GRFP DGE-1122492 | Ann Feke |
| National Institutes of Health | T32 GM007499 | Ann Feke |

The funders had no role in study design, data collection and interpretation, or the decision to submit the work for publication.

## Author contributions

Ann Feke, Conceptualization, Resources, Data curation, Software, Formal analysis, Funding acquisition, Validation, Investigation, Visualization, Methodology, Writing—original draft, Writing—review and editing, Conception, design, and creation of decoy library; Wei Liu, Chin-Mei Lee, Conceptualization, Resources, Funding acquisition, Conception, design, and creation of decoy library; Jing Hong, Resources, Investigation, Creation of decoy library; Man-Wah Li, Conceptualization, Resources, Conception, design, and creation of decoy library; Elton K Zhou, Resources, Creation of decoy library; Joshua M Gendron, Conceptualization, Supervision, Funding acquisition, Methodology, Project administration, Writing—review and editing, Conception and design of decoy library

## Author ORCIDs
Ann Feke http://orcid.org/0000-0002-8246-0056
Chin-Mei Lee http://orcid.org/0000-0003-3870-4268
Joshua M Gendron http://orcid.org/0000-0001-8605-3047

### Decision letter and Author response
Decision letter https://doi.org/10.7554/eLife.44558.034
Author response https://doi.org/10.7554/eLife.44558.035

## Additional files

### Supplementary files
• Source code 1. MATLAB script called by outlier_analysis that determines the number of subpopulations within a dataset.
DOI: https://doi.org/10.7554/eLife.44558.025

• Source code 2. MATLAB script called by batch_outlier to compare individual datasets to the control dataset.
DOI: https://doi.org/10.7554/eLife.44558.026

• Source code 3. MATLAB script used to batch-process circadian imaging data.
DOI: https://doi.org/10.7554/eLife.44558.027

• Supplementary file 1. All generated data and publications which reference genes in our decoy library.
DOI: https://doi.org/10.7554/eLife.44558.028

• Supplementary file 2. IP-MS results from the MAC3B decoy.
DOI: https://doi.org/10.7554/eLife.44558.029

• Supplementary file 3. Primers used in this Study. (*Czechowski et al., 2004*; *Farré et al., 2005*; *Lee and Thomashow, 2012*)
DOI: https://doi.org/10.7554/eLife.44558.030

• Supplementary file 4. ABRC Stock numbers for pENTR-decoy vectors, 35S:HIS-FLAG-decoy vectors, and 35S:HIS-FLAG-decoy transgenic seed stocks.
DOI: https://doi.org/10.7554/eLife.44558.031

• Transparent reporting form
DOI: https://doi.org/10.7554/eLife.44558.032

### Data availability
All data generated or analyzed during this study are included in the manuscript and supporting files. Source data is provided for Figure 2, Figure 2 Supplement 1, Figure 3, Figure 3 Supplement 1, Figure 4, Figure 4 Supplement 1, Figure 5, Figure 5 Supplement 1, Figure 6, Figure 6 Supplement 1, Figure 8, and Figure 8 Supplement 1.

The following datasets were generated:

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
