## [Decision Letter]

Thank you for submitting your article "Decoys provide a scalable platform for genetic analysis of plant E3 ubiquitin ligases that regulate circadian function" for consideration by *eLife*. Your article has been reviewed by three peer reviewers, and the evaluation has been overseen by a Reviewing Editor and Christian Hardtke as the Senior Editor. The reviewers have opted to remain anonymous.

The reviewers have discussed the reviews with one another and the Reviewing Editor has drafted this decision to help you prepare a revised submission.

The reviewers appreciated that your decoy approach (employing inactive variants) may enable the field to identify the molecular function of E3 ligases. The manuscript currently provides only modest biological advances, but the reviewers agreed that this reverse genetic approach could overcome genetic redundancy, which will help to push the field further (see reviewer comments below). The reviewers were largely positive; however, they raised substantial concerns on how this resource could be used. Hence, it will be mandatory to define how the plants or "lines" would be provided and how precisely the re-screening may look like. Would a lab that wanted to examine phenotypes in the decoy lines have to generate new transgenics, or would the existing lines (which generation) be appropriate for re-screening? Where will the authors deposit the constructs and plant lines?

The reviewers moreover advised the authors to balance the discussion on the currently provided biological insights (see more specific reviewer comments below).

*Reviewer #1:*

In the manuscript "Decoys provide a scalable platform for genetic analysis of plant E3 ubiquitin ligases that regulate circadian function" Feke and colleagues introduce a toolset intended to allow the genetic analyses of E3 ligase function in circadian rhythm.

In this study, authors employed inactive variants to test genetic function and overcome redundancy. Proof-of-principle was provided by the authors in a previous publication, in which they analysed the function of three related F-boxes that are involved in the regulation of the circadian rhythm (Lee et al., 2018). The idea behind the approach is that the overexpressed inactive E3 variants will bind to their substrate(s), and thus, although not shown, protect it from ubiquitination and potentially degradation, by outcompeting the endogenous E3s.

For the screen a large set of F-box proteins and U-box type E3 ligases were cloned and overexpressed in Col-0 plants carrying the CCA1prom:Luc reporter construct. The resulting T1 lines were then analysed for changes in phase and period of the circadian rhythm.

As a result authors were able to identify various E3s with a potential function in the circadian rhythm, of which one F-Box and one PUB had a major effect.

Overall the work is of high quality and the authors use various controls and stringent analysis conditions. The manuscript is well written, although entire sections of the Results are actually discussion and do not include any results.

However, use of inactive variants (decoys) for functional analyse is not uncommon and similar approaches have been reported previously, as also cited, and includes work by the authors themselves. The novelty in this manuscript therefore lies within the scale of the approach to carry out a screen.

The authors provide some results that suggest that the identified U-box proteins Prp19A and B (PUB59 and PUB60) may influence the circadian rhythm. Prp19 is an essential components of the Nineteen Complex (NTC) that performs pre-mRNA splicing and is highly conserved in all eukaryotes. The pub59 pub60 double mutants are shown to display defects in circadian rhythm based on analysis of CCA1 transcription, and lines carrying the CCA1prom:Luc reporter construct expressing PUB60 show changes in phase. However, because PUB59 and PUB60 are expected to be essential for splicing, it is not surprising that transcription is affected. Importantly, authors do not provide data that unlinks the putative function of PUB59 and PUB60 in the circadian rhythm, from an indirect effect due to defects in splicing. As a consequence it is also unclear, whether the screen was successful.

*Reviewer #2:*

In this manuscript, Feke et al. describe the generation of a population of 'decoy' E3 ubiquitin ligase constructs for overexpression of versions of these proteins predicted to bind their substrates but not the ubiquitination machinery (covering ~1/4 of the E3 ligases in Arabidopsis). They screen Arabidopsis plants transgenic for these constructs (~220 constructs tested) for altered expression of a clock-regulated luciferase reporter and identify a number of constructs that alter circadian period and/or phase. They follow up on the strongest of these, which they call PUB60 and which causes a long period phenotype. PUB60 encodes an Arabidopsis homolog of PRP19, and has previously been identified as MAC3B. Feke et al. report that plants double mutant for PUB60 and its close homolog PRP59 have long-period circadian phenotypes, consistent with the PUB60 decoy acting as a dominant negative. Consistent with previous studies in Arabidopsis (Monaghan et al., 2009; Jia et al., 2017), they find additional components of the PRP19 complex co-purifying with their PUB60 decoy protein. Finally, they find RNA splicing of the clock transcript PRR9 is altered in *pub59 pub60* mutants. Wang et al., 2012, previously reported increased peak levels of isoforms *PRR9b* and *PRR9c* in plants mutant for SKIP, which encodes a component of the PRP19 complex. Although the data are noisy, Feke et al. find decreased peak and trough levels of *PRR9b* and decreased trough levels of *PRR9c* in *pub59 pub60* mutants. They suggest the clock phenotype of *pub59 pub60* mutants may be due in part to missplicing of PRR9.

Although the new biological insights provided in this manuscript are modest, the authors have generated tools that might prove useful to many plant biologists. The reverse genetic approach described here may indeed allow the identification of E3 ubiquitin ligase genes involved in specific processes difficult to identify by forward genetic approaches given genetic redundancy. However, the authors have not provided enough information to allow me to determine how broadly useful these tools may be. More information is required regarding the screening protocol and the biological resources available to the community.

What generation of transgenic plants was screened; T1? (The authors state "A decoy 'line' is defined as a single, independent T1 insertion line". If T1 plants were screened, it would be more accurate to refer to the screened organisms as 'plants' or 'transformants' rather than 'lines'.)

Additional information on the transgenic seeds that would be made available to the community is also needed to assess their general utility. What generation seed stocks would be available to the community? If each point in each plot does represent data from a T1 individual, it is not clear why 'subpopulation' phenotypes arise for some constructs. How stable are these overexpressing lines? How many transformant lines would other users need to assess for each construct? Finally, will seeds and constructs be deposited in the ABRC? Will MTA be required to obtain the constructs and seeds? If the biological materials available are frequently heterogeneous, it is hard to see how they would be broadly useful.

*Reviewer #3:*

In the submitted manuscript, Feke and colleagues describe a new screening platform to overcome traditional genetic challenges and discover putative redundant E3 ligases implicated in protein degradation.

The authors generated a library of transgenic *Arabidopsis thaliana* lines expressing dominant-negative variants of E3 ligases. This resource will be key for the community as finding E3 ligases is generally not easy due to the transient interaction between the enzyme and its substrate(s).

As a proof of concept, Feke and colleagues identified new ubiquitin ligases that control the circadian clock in Arabidopsis. They overexpressed tagged decoy E3 ligase lines using the strong CaMV 35S promoter into plants already expressing Luciferase under the control of the CCA1 promoter. Such decoy lines are able to interact with their substrate but cannot recruit the E2 ubiquitin conjugating enzyme thereby preventing protein degradation. In their screen, the authors uncovered several E3 ligases but focused their attention on PUB59 and 60, two U-box proteins. They showed that redundant PUB59/60 control splicing of PRR9, a circadian clock gene.

This manuscript represents a milestone in the study of E3 ligases in Arabidopsis. Indeed, there are more than 700 F-box proteins in the Arabidopsis genome and only a few of them have been identified using traditional genetic screens. As such, this study provides a powerful platform for better understanding of the role of E3 ligases in developmental processes.

---

## [Author Response]

Reviewer #1:[…] Overall the work is of high quality and the authors use various controls and stringent analysis conditions. The manuscript is well written, although entire sections of the Results are actually discussion and do not include any results.

Thank you for your comments on our manuscript. Internally, we debated the placement of the supplementary gene information about each potential hit from the screen. Initially, we included this information in the Discussion as suggested, but were unsure about the placement of data that has been mined from publicly available datasets. We ultimately came to the conclusion that because we were mining and compiling publicly available data, creating tables (Tables 1-3), and drawing new conclusions, the information belongs in the Results section. We hope that this explains our decision, but if we are mistaken in our assessment, we will gladly move these sections to the Discussion.

However, use of inactive variants (decoys) for functional analyse is not uncommon and similar approaches have been reported previously, as also cited, and includes work by the authors themselves. The novelty in this manuscript therefore lies within the scale of the approach to carry out a screen.The authors provide some results that suggest that the identified U-box proteins Prp19A and B (PUB59 and PUB60) may influence the circadian rhythm. Prp19 is an essential components of the Nineteen Complex (NTC) that performs pre-mRNA splicing and is highly conserved in all eukaryotes. The pub59 pub60 double mutants are shown to display defects in circadian rhythm based on analysis of CCA1 transcription, and lines carrying the CCA1prom:Luc reporter construct expressing PUB60 show changes in phase. However, because PUB59 and PUB60 are expected to be essential for splicing, it is not surprising that transcription is affected. Importantly, authors do not provide data that unlinks the putative function of PUB59 and PUB60 in the circadian rhythm, from an indirect effect due to defects in splicing. As a consequence it is also unclear, whether the screen was successful.

We apologize that the goals of the work were not made sufficiently clear. We designed the screen to identify any E3 ligases involved in circadian clock function, whether controlling degradative or regulatory ubiquitylation. The major advantage of our technique is its ability to overcome genetic redundancy. MAC3A/3B (PUB59/PUB60) were an ideal choice for follow-up studies because there are clearly only two potentially redundant copies of the gene. We believe the success of the screen is clear because our technique overcame redundancy to identify new regulators of the clock.

We do agree that the distinction between degradative and regulatory ubiquitylation can be made more explicit. To clarify, we have included additional description of the functions of E3 ubiquitin ligases in both degradative and regulatory ubiquitylation events (Introduction, third paragraph). We hope this supports the idea that our screen is unbiased in this manner.

Reviewer #2:[…] Although the new biological insights provided in this manuscript are modest, the authors have generated tools that might prove useful to many plant biologists. The reverse genetic approach described here may indeed allow the identification of E3 ubiquitin ligase genes involved in specific processes difficult to identify by forward genetic approaches given genetic redundancy. However, the authors have not provided enough information to allow me to determine how broadly useful these tools may be. More information is required regarding the screening protocol and the biological resources available to the community.

Thank you for this comment. We added an extensive section to the Discussion detailing the potential applications of the decoy library and the available reagents. We feel that by adding this section it dramatically improves the quality of the manuscript by giving the readers a sense of how important this resource can be for the scientific community. The section is titled “Usage of the Decoy Library”. Furthermore, we have added an additional section to the Materials and methods titled “Resource Distribution” to describe how the materials will be made available to the public.

What generation of transgenic plants was screened; T1? (The authors state "A decoy 'line' is defined as a single, independent T1 insertion line". If T1 plants were screened, it would be more accurate to refer to the screened organisms as 'plants' or 'transformants' rather than 'lines'.)

Thank you for the suggestion. We realize that the use of the term “line” to mean any individual T1 plant was confusing. We have substituted the term “line” for “plant” in our manuscript.

Additional information on the transgenic seeds that would be made available to the community is also needed to assess their general utility.

We have added an extensive section detailing the materials available to the scientific community (Discussion section “Usage of the Decoy Library”). Additionally, we have added a section detailing the distribution to the Materials and methods section (“Resource distribution”).

What generation seed stocks would be available to the community?

Although we recommend re-transforming and screening T1 transgenic plants, the library will be available as pooled T2 transgenics that originated from approximately 20 T1 transgenic parents.

If each point in each plot does represent data from a T1 individual, it is not clear why 'subpopulation' phenotypes arise for some constructs.

Although there could be multiple ways that these subpopulations could arise, the most likely is the variation in the genomic context of the insertion site. This would likely lead to differences in expression of the transgene, and correspondingly variations in phenotype. We provide an in-depth discussion of this in a previous paper (Lee and Feke, et al.,2018).

How stable are these overexpressing lines?

In our lab, we have maintained multiple transgenic lines in the Col-0 background for numerous generations without loss of protein expression. For this reason, we believe that these overexpression lines are equivalent to any overexpression line.

How many transformant lines would other users need to assess for each construct?

In our screening protocol, we were able to screen 20 T1 transgenics in a high throughput manner, although this was limited by experimental constraints. With this number we were able to identify transgenic populations that contain subpopulations of three or more individuals. For this reason we recommend an equivalent number, although the experimental constraints will drive the number of transgenics that can be analyzed.

Finally, will seeds and constructs be deposited in the ABRC? Will MTA be required to obtain the constructs and seeds?

We will be depositing the seeds and constructs in ABRC. We do not foresee the requirement of an MTA.

If the biological materials available are frequently heterogeneous, it is hard to see how they would be broadly useful.

Thank you for this comment. The individual decoy transgenics are not heterogeneous, but there is heterogeneity at the population level, as would be expected for any transgenic study. We believe the population-level heterogeneity serves as a strength, not as a weakness. It is recommended, if not required, that multiple independent insertion transgenic lines are analyzed to avoid the lack of reproducibility caused by analysis of single transgenic lines. Our data in this manuscript, and previously (Lee and Feke et al.,2018), makes a strong case for taking a population-level approach towards the analysis of overexpression lines when possible. To expand on this, we have included a deeper discussion of this idea in our manuscript in the section “T2 transgenic Col-0 seeds”, and we have provided the reagents for users to do individual or population-level studies.